# Calibrated sea level contribution from the Amundsen Sea sector, West Antarctica, under RCP8.5 and Paris 2C scenarios

Sebastian H. R. Rosier[1,2], G. Hilmar Gudmundsson[1], Adrian Jenkins[1], and Kaitlin A. Naughten[3]

[1]Northumbria University, Newcastle, UK
[2]University of Zurich, Zurich, Switzerland
[3]British Antarctic Survey, Cambridge, UK

**Correspondence:** Sebastian Rosier (sebastian.rosier@northumbria.ac.uk)

**Abstract.** The Amundsen Sea region in Antarctica is a critical area for understanding future sea level rise due to its rapidly changing ice dynamics and significant contributions to global ice mass loss. Projections of sea level rise from this region are essential for anticipating the impacts on coastal communities and for developing adaptive strategies in response to climate change. Despite this region being the focus of intensive research over recent years, dynamic ice loss from West Antarctica and in particular the glaciers of the Amundsen Sea represent a major source of uncertainty for global sea level rise projections. In this study, we use ice sheet model simulations to make sea level rise projections to the year 2100 and quantify the associated uncertainty using a comprehensive Bayesian approach aided by deep surrogates. The model is forced by climate and ocean model simulations for the RCP8.5 and Paris2C scenarios, and is carefully calibrated using measurements from the observational period. We find very similar sea level rise contributions of $19.0\pm 2.2$ mm and $18.9\pm 2.7$ mm by 2100 for Paris2C and RCP8.5 scenarios, respectively. A subset of these simulations, extended to 2250, show an increase in the rate of sea level rise contribution and clearer differences emerge between scenarios, with increasing snow accumulation in RCP8.5 resulting in less cumulative mass loss. Our model simulations include both a cliff-height and hydrofracture driven calving processes and yet we find no evidence of the onset of rapid retreat that might be indicative of an unstable calving front retreat in any simulations within our modelled timeframe.

## 1 Introduction

The Amundsen Sea region in Antarctica plays a pivotal role in influencing global sea levels, and the dynamics of its glaciers have garnered significant attention in the context of climate change. Since the advent of satellite observations, glaciers in this region, in particular Pine Island and Thwaites glaciers, have undergone periods of extensive thinning and retreat, believed to be largely driven by increased ocean melting of their floating ice shelves (Pritchard et al., 2012; Paolo et al., 2015; Gudmundsson et al., 2012; Jenkins et al., 2018). While the region already accounts for a substantial portion of current rates of global sea level rise (SLR), vulnerability to unstable retreat might lead to greatly increased rates of ice loss (Davison et al., 2023; Rosier et al., 2021; Pollard et al., 2015; Feldmann and Levermann, 2015). Modelling challenges, most notably the representation of ocean forcing, model initialisation and the presence of possible future tipping points, mean that there is a wide spread in SLR

projections (Robel et al., 2019) and the latest IPCC special report emphasises the "deep uncertainty" associated with ice sheet model projections (Pörtner et al., 2019).

Existing SLR projections for the ASE vary widely, with high-end scenarios suggesting contributions of $\sim$0.3 m by 2200 under strong warming pathways (DeConto et al., 2021). The relative SLR contribution of Pine Island and Thwaites glaciers has varied considerably over the historical record but recent research has focused increasingly on Thwaites glacier due to its vulnerable configuration and greater SLR potential. Past studies have also identified tipping points where retreat past certain thresholds could trigger irreversible loss via marine ice sheet instability (Rosier et al., 2021), potentially leading to eventual collapse of much of the West Antarctic Ice Sheet (Feldmann and Levermann, 2015). Due to its complexity and importance, the ASE region has attracted a lot of research interest in the past decade and we include a more complete overview of previous modelling studies in Sect. 6.

A major challenge in modelling the response of Antarctic glaciers to future emissions scenarios lies in the representation of ocean processes within ice sheet models. Generally, ice sheet modelling studies use simple parameterisations of basal melting, however, this approach cannot hope to capture the complex response of the ocean and ice shelf cavities to atmospheric forcing. This is particularly true in the ASE, where increased ice flux through basal melting has not been driven by some simple increasing trend in ambient ocean temperature, but rather intermittent changes in thermocline depth, driven by complex non-local processes, leading to a spatially and temporally heterogeneous response in ice shelf melting (Holland et al., 2022; Jenkins et al., 2018). Increasingly, coupled ice-sheet/ocean models are becoming available that address this shortcoming (e.g. De Rydt and Naughten, 2024; Bett et al., 2024), however these come with a computational cost that makes running large numbers of simulations impractical. This is currently a major downside of the coupled modelling approach, because being able to run a large number of simulations is necessary in order to be able to properly explore the potentially large uncertainty associated with a model prediction.

Here, we use an alternative approach by assuming that feedbacks between the ice sheet and ocean are sufficiently small, over the timescales of interest, that we can use standalone ocean model simulations to capture the response of ice shelf cavities to atmospheric forcing. We model melt rates with an adaptation of the 2D plume model of Lazeroms et al. (2018) that is driven by a stratified water column whose properties in each cavity are obtained from recent state-of-the-art model simulations of the region (Naughten et al., 2023). Both the ocean and ice sheet models are driven by atmospheric forcing from CESM1 for two emissions scenarios: RCP8.5 and Paris2C. We use a comprehensive Bayesian framework to first calibrate our ice sheet model using observations and then evaluate the uncertainty arising from model parameters and internal climate variability, leading to SLR projections for the ASE region with corresponding uncertainty estimates.

## 2  Data and Methods

In order to arrive at our final estimated sea level contribution and calculate the associated uncertainties, a number of different modelling components and methods need to come together. These can broadly be divided into two main steps: (1) outputs from a large ensemble of forward model simulations for the period 1st January 1996 to 1st January 2021 (Úa-*obs*) are compared to

| parameter | name | model component | prior range |
|---|---|---|---|
| $m$ | sliding exponent | ice flow | 2 to 8 |
| $n$ | creep exponent | ice flow | 2 to 5 |
| $p$ | precipitation factor | SMB | 0.03 to 0.15 |
| $\sigma_T$ | temperature variability | SMB | -0.3277 to -0.3821 |
| $C_d^{1/2}\Gamma_{TS}$ | Stanton number | BMB | $6.5 \times 10^{-5}$ to $4.6 \times 10^{-4}$ |
| $E_0$ | plume entrainment | BMB | $7.9 \times 10^{-3}$ to $6.2 \times 10^{-2}$ |
| $\dot{h}_e$ | thickness change error | inversion | $3 \times 10^{-3}$ to $3 \times 10^3$ |
| $\gamma_{sC}$ | $C$ smoothness penalty | inversion | $1.3 \times 10^3$ to $1.3 \times 10^6$ |
| $\gamma_{sA}$ | $A$ smoothness penalty | inversion | 0.25 to 250 |
| $\gamma_{aC}$ | $C$ prior penalty | inversion | $5 \times 10^{-2}$ to $5 \times 10^5$ |
| $\gamma_{aA}$ | $A$ prior penalty | inversion | 0.25 to 250 |
| $C_Q$ | calving runoff factor | calving | 0 to 200 |

**Table 1.** Uncertain model parameters included in the Bayesian calibration step described in Section 2.4. Note that range refers to the minimum and maximum values for parameters with uniform priors and the $\pm 3\sigma$ range for variables with Gaussian priors.

observations over the same period to arrive at a likelihood which, given priors for our model parameters, enables us to calculate posterior probabilities for uncertain model parameters. (2) A second set of model simulations (Úa-*fwd*) use the calculated model parameter distributions to predict changes in ice volume between 1st January 2021 and 1st January 2100 for different emissions scenarios and quantify the associated uncertainty. Since the forward model simulations are computationally expensive, the uncertainty quantification steps in (1) and (2) are performed on surrogates of the forward model, as described in more detail below. An overview of our methodology is presented in Fig. 1. In the description that follows, we only distinguish between the two sets of forward model simulations where they necessarily differ, e.g. in the model forcing. Where a difference is not stated explicitly, the details are the same. In Section 2.1 we give an overview of the ice-flow model, with a focus on highlighting the uncertain model parameters. These uncertain model parameters, related to ice flow, surface mass balance (SMB), basal mass balance (BMB), calving and model initialisation, are listed in Table 1. Note that parameters related to surface mass balance and calving are not included in the Bayesian calibration step, as explained in Sec. 2.4 and App. C. Section 2.2 summarises the external forcing for all the simulations, Sec. 2.3 presents the observations used to constrain our model parameters and Sec. 2.4 presents the model calibration methodology.

## 2.1 Ice flow model

All simulations use the Úa community ice-flow model (Gudmundsson, 2020, 2024), which solves the dynamical equations for ice flow with the shallow ice stream approximation (SSTREAM or SSA; Hutter 1983; MacAyeal 1989), using the finite element method on an irregular triangular mesh. The momentum and mass conservation equations are solved simultaneously using a

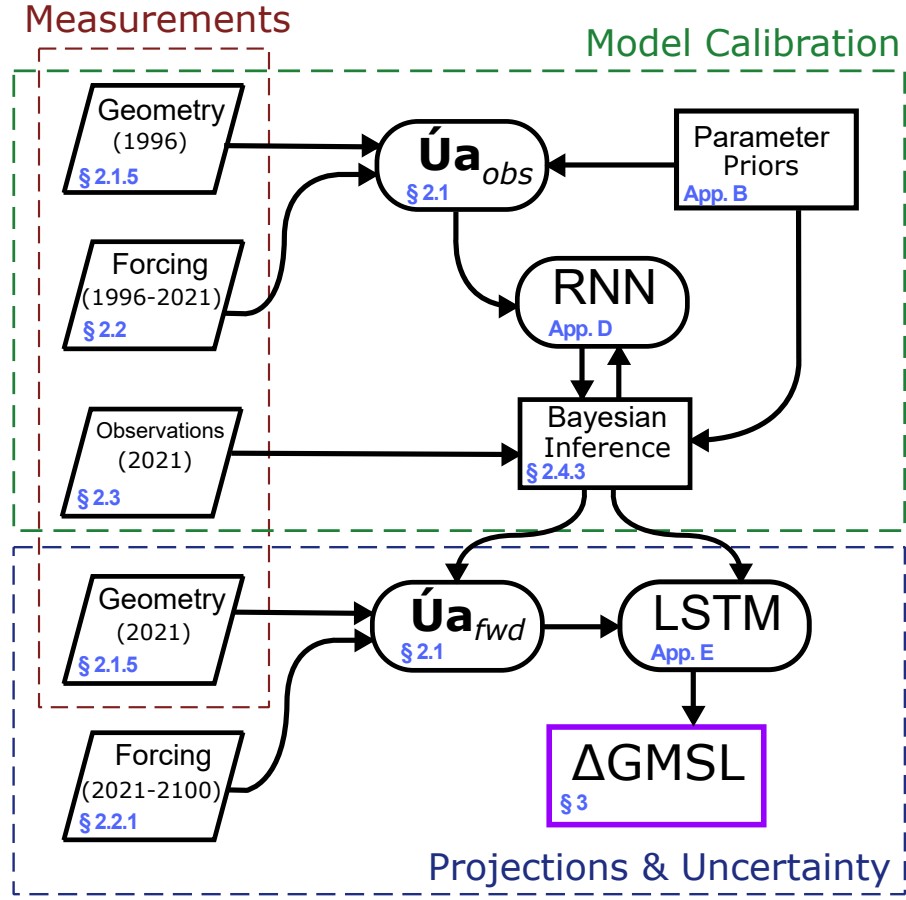

**Figure 1.** Schematic overview of the inputs, models and processing steps that lead to projections of sea level contribution from the ASE up to the year 2100. GMSL stands for global mean sea level, RNN for recurrent neural network and LSTM refers to a long short-term memory type network. We distinguish model calibration steps, resulting in posterior probability distributions for uncertain parameters (dashed green box) from projection/uncertainty steps (dashed blue box). Section or appendices numbers for the relevant step are given in the lower left of each box.

fully implicit forward time integration with respect to both velocities and thickness. Úa uses inverse methods to estimate the

uncertain spatial distributions of the rate factor $A$ and the basal slipperiness $C$, as described in Section 2.1.4. The rate factor determines the relationship between the strain rates $\dot{\varepsilon}$ and deviatoric stresses $\tau$ through the Glen–Steinemann flow law (Glen, 1955):

$$\dot{\varepsilon} = A \tau_E^{n-1} \tau \tag{1}$$

where $n$ determines the non-linearity of this constitutive law and we treat as an uncertain model parameter (Millstein et al.,

2022). The relationship between basal traction, $\boldsymbol{t}_b$, and basal sliding velocity $\boldsymbol{v}_b$ is given by a mixed Coulomb-Weertman sliding

law of the form:

$$t_b = \frac{\mu_k N}{\mu_k N + \beta^2 \|v_b\|} \beta^2 v_b \,, \tag{2}$$

where $\beta^2 = C^{-1/m} \|v_b\|^{1/m-1}$, $N$ is the effective pressure, $\mu_k$ is the coefficient of friction, and $m$ is an uncertain model parameter that controls the nonlinearity of the sliding law in the Weertman term. The sliding law (2) is arrived at by setting the basal traction, $t_b$ equal to the reciprocal sum of the Weertman, $t_b^W$ and Coulomb tractions, $t_b^C$ as follows

$$\frac{1}{\|t_b\|} = \frac{1}{\|t_b^W\|} + \frac{1}{\|t_b^C\|} \,, \tag{3}$$

where

$$\|t_b^W\| = \beta^2 \|v_b\|, \tag{4}$$

$$\|t_b^C\| = \mu_k N \,. \tag{5}$$

This sliding law has been used previously in Barnes and Gudmundsson (2022) and has been implemented in the main Úa branch since 2020, along with the Weertman, Tsai, Coulomb and Budd sliding laws.

We use the simplification that the effective pressure is the difference between ice overburden pressure and water pressure, assuming perfect hydraulic conductivity with the ocean. This results in $N = 0$ zero at the grounding line and $N$ increasing inland as a function of ice thickness. In the limit, as $N \to 0$ basal sliding follows purely Coulomb behaviour, whereas as $N \to \infty$ basal sliding is purely Weertman. The use of a mixed sliding law such as this one has become more frequently used by the community in recent years due to its flexibility in representing both plastic and hard bed sliding, and an improved representation of conditions near the grounding line (e.g. Gudmundsson et al., 2023; Hill et al., 2024).

### 2.1.1 Surface mass balance

We use a Positive Degree Day (PDD) model to capture two main processes: (1) changes in precipitation resulting from large adjustments of the ice sheet surface elevation and (2) production of runoff that contributes to both surface mass balance and can impact the calving rate (2.1.3). Note that other factors influencing changes in precipitation, such as those driven by temperature variations, are already accounted for in the CESM1 climate model, which provides our atmospheric forcing; therefore, these factors are not incorporated into the PDD model. With this modification, the remaining formulation of our PDD model broadly follows the descriptions given in Huybrechts and de Wolde (1999) and Janssens and Huybrechts (2000). Precipitation ($P$) is given by

$$P = P_{\text{cesm}} e^{p \Delta T} \,, \tag{6}$$

where $P_{cesm}$ is the bias corrected (see Sect. 2.2) precipitation forcing from the CESM1 model and $\Delta T = \gamma_p(s - s_{\text{init}})$ where $\gamma_p$ is the lapse rate and $s_{\text{init}}$ is the initial ice sheet surface elevation. The parameter $p$ captures the expected increase in snowfall arising from the increased moisture content of warmer air, as suggested by climate models (Aschwanden et al., 2019), and is treated as an unknown within our Bayesian framework.

Air temperature is modelled with a normal distribution whose mean is obtained from bias corrected output from CESM1 ($T_{\text{cesm}}$) and standard deviation ($\sigma_M$) is given by

$$\sigma_M = \sigma_T T_{\text{cesm}} + 1.66, \tag{7}$$

where $\sigma_T$ is treated as an unknown parameter. This formulation for $\sigma_M$ was chosen as it has been shown to provide a better fit to observations than using a constant standard deviation (Wake and Marshall, 2015), a result that was confirmed by our own testing. With this simple statistical model of air temperature we can calculate the proportion of precipitation falling as rain, with the remainder therefore falling as snow. This snow accumulates and can be melted as a function of $\gamma_{\text{snow}} PDD$, where $PDD$ is the number of positive degree days and $\gamma_{\text{snow}}$ is the degree day factor for snow. Any meltwater initially percolates down and refreezes as superimposed ice. If the amount of superimposed ice exceeds a threshold, no more superimposed ice is formed and the meltwater contributes to a runoff term. We use the capillary retention model of Janssens and Huybrechts (2000) to calculate this threshold, which takes into account both the refreezing process and the capillary suction effect of the snowpack. If all the snow is melted, the superimposed ice layer is melted (via a degree day factor for ice $\gamma_{\text{ice}}$) and finally if that is completely removed the glacier ice itself can be melted, both of which also contribute to a runoff term. Production of runoff is kept track of locally and is subsequently used in our calving law (2.1.3) but we do not model the routing of meltwater across the ice shelf. Our PDD model is updated in monthly time increments in a separate time-stepping scheme to the ice sheet model (whose time steps adaptively change based on convergence).

### 2.1.2 Basal mass balance

Basal melting or ablation of ice at its base in these model simulations occurs as a result of transfer of heat from the ocean (on floating ice shelves) and frictional heating where ice is in contact with the bed (i.e. grounded). Ocean induced basal melting accounts for over half of the mass loss of the AIS (Rignot et al., 2013) and is believed to be responsible for much of the measured changes during the observational period, so this component of the model requires particularly careful treatment. Complex feedbacks between ice shelf geometry, cavity circulation, atmospheric forcing and conditions at the domain boundary mean that accurately predicting melt rates is a considerable challenge and generally requires computationally expensive general circulation models, an option that is not feasible for our purposes. Instead, we use an adaptation of the 2D plume model presented by Lazeroms et al. (2018, 2019) and originally proposed by Jenkins (1991, 2011). This retains much more physics than simpler but commonly used depth-dependent parameterisations, while remaining computationally inexpensive to run. A detailed description of our modification to the implementation of Lazeroms et al. (2018, 2019) can be found in App. A and what follows below is only a brief overview introducing the most relevant concepts and parameters.

The plume model considers a two layer system in which an upper layer that is freshened by the addition of meltwater travels along the ice shelf base, driven by a buoyancy difference relative to the ambient ocean layer beneath that has temperature $T_a$ and salinity $S_a$. The plume itself has temperature $T_p$, salinity $S_p$, thickness $D_p$ and velocity $U_p$ and travels in a coordinate system orientated with $X$ along the ice-shelf base of slope $\alpha$ such that $X = 0$ at the grounding line. The plume equations result from conservation of mass, momentum, heat and salt within the plume and are detailed in Lazeroms et al. (2019). As the plume

travels in a positive $X$ direction its volume is increased through entrainment of ambient ocean water ($\dot{e}$) and a meltwater flux at the ice-ocean interface ($\dot{m}$). The entrainment rate is given by

$$\dot{e} = E_0 U_p \sin\alpha \tag{8}$$

where $E_0$ is an uncertain model parameter. At the ice-ocean interface, melting arises from a balance between turbulent exchange of heat to the ice/ocean interface and latent heat of fusion in the ice:

$$\left(\frac{L}{c}\right)\dot{m} = C_d^{1/2}\Gamma_{TS}U_p(T_p - T_f), \tag{9}$$

where $L$ is the latent heat of fusion in ice, $c$ is the specific heat capacity of water and the freezing temperature is a function of the plume salinity and the depth of the ice shelf base

$$T_f = \lambda_1 S_p + \lambda_2 + \lambda_3 b, \tag{10}$$

where $\lambda_1$, $\lambda_2$ and $\lambda_3$ are constant parameters. The thermal Stanton number, $C_d^{1/2}\Gamma_{TS}$ is an uncertain model parameter that plays a key role in determining how effectively warm ocean temperatures can melt the ice-shelf. While the original plume model of Jenkins (1991) solved a system of first order differential equations to obtain the values of $U_p$ and $(T_p - T_f)$ needed to calculate the melt rate, Lazeroms et al. (2019) derived analytical expressions for those variables as a function of distance from the grounding line (detailed in Appendix A2), which we use in this study.

Beneath grounded ice, we calculate the frictional heating term resulting from basal sliding, which in areas of high basal sliding velocity can become non-negligible. We do not consider the contribution of geothermal heat flux since this term is more important for the ice sheet thermal state, which is inferred via our inverse approach, and in fast-flowing regions produces negligible melting compared to the frictional heating term. We calculate melting resulting from this frictional heating as

$$a_b = \frac{\boldsymbol{t}_b \cdot \boldsymbol{v}_b}{\rho L} \tag{11}$$

where $\boldsymbol{t}_b$ and $\boldsymbol{v}_b$ are the basal traction and basal velocity, respectively. The resulting melt rates in the fastest flowing regions are on the order of $1\ \mathrm{myr}^{-1}$ which is two orders of magnitude larger than basin-wide averaged basal melting due to geothermal heat flux (Joughin et al., 2009).

### 2.1.3 Calving Law

We allow calving fronts to evolve dynamically during our simulations, using a calving law that depends on a parameterisation of local crevasse depth, as presented in Pollard et al. (2015); DeConto et al. (2021). Total crevasse depth $d_{tot}$ is a sum of terms depending on flow divergence, accumulated strain, ice thickness and surface water. The overall calving rate is given by

$$R_c = 3000\max[0, \min[1, (r - r_c)/(1 - r_c)]] \tag{12}$$

where $r = d_{tot}/h$ is the proportion of crevasse depth to total ice thickness and $r_c = 0.75$ is a critical value suggested by Pollard et al. (2015), i.e. calving occurs when a crevasse penetrates through 75% of the ice thickness. Our primary motivation here is

to include the possibility that a large increase in surface meltwater could lead to hydrofracturing and precipitate the breakup of ice shelves in the region. To this end, we explore the sensitivity of the calving law to the presence of surface meltwater via the parameter $C_Q$, such that contribution to the total crevasse depth is $f_r C_Q R^2$ (DeConto et al., 2021). Here $f_r$ is a factor that scales from 0 to a maximum of 1 for areas of low to moderate meltwater production (0 to 1.5 myr$^{-1}$). Following DeConto et al. (2021), we use a range for $C_Q$ of 0 to 200 m$^{-1}$yr$^{-2}$.

In addition to the calving law in Eq. 12, and also following Pollard et al. (2015), we implement structural failure of ice cliffs necessary to allow for the possibility of a MICI (Marine Ice Cliff Instability) driven retreat. The cliff-based calving rate ramps up linearly from zero at a cliff height of 80 m to 3 kmyr$^{-1}$ at a cliff height of 100 m. Our resulting final calving rate, defined everywhere as a field for the level-set method, is then calculated as the maximum of either this cliff-based calving rate or the calving rate given by Eq. 12.

For a calving front to remain fixed, the calving rate must equal the ice velocity normal to the calving front. Generally, for a given calving law, those two quantities are not equal, resulting in migrating calving fronts over time. The Úa model uses the level-set method to solve for this moving boundary problem. The key idea is to introduce a new field variable, $\varphi(x, y, t)$, and define the calving front as the set of points in the $(x, y)$ plane for which $\varphi(x, y, t) = 0$ for any time $t$. After initial initialisation at the start of the run, the level set function ($\varphi$) is evolved by solving the level set equation

$$\partial_t \varphi + F \|\nabla \varphi\| = 0, \tag{13}$$

where the scalar $F$ is the speed in outward normal direction, $\hat{\boldsymbol{n}}$, to the calving front, i.e.

$$F = \boldsymbol{v} \cdot \hat{\boldsymbol{n}} - c, \tag{}$$

where $\boldsymbol{v}$ is the material velocity and $c$ the calving rate. A calving law provides $c$ as a function of some other variables such as cliff height, state of stress, etc., for example as $c = c(f, \sigma)$, where $f$ is the cliff height and $\sigma$ the Cauchy stress tensor. For the calving law to be physically admissible, the function $f$ must be frame invariant, a condition that, as shown in Mitcham and Gudmundsson (2022), is not satisfied by all previously proposed calving laws.

The implementation details are provided in the Úa Compendium (Gudmundsson, 2024). A similar approach is used in the Ice-sheet and Sea-level System Model (ISSM), and further technical details for both the ISSM and the Úa ice sheet models, are provided in Cheng et al. (2023).

### 2.1.4 Model Initialisation

Úa uses the adjoint method to invert for $C$ and $A$ using observed surface ice velocity and the rate of change of ice thickness. It minimises the cost function $J = I + R$ where

$$R = \frac{1}{2\mathcal{A}} \int \left( \gamma_{sC}^2 \left[ \nabla \log_{10}(C/\tilde{C}) \right]^2 + \gamma_{aC}^2 \log_{10}(C/\tilde{C})^2 \right) d\mathcal{A} + \frac{1}{2\mathcal{A}} \int \left( \gamma_{sA}^2 \left[ \nabla \log_{10}(A/\tilde{A}) \right]^2 + \gamma_{aA}^2 \log_{10}(A/\tilde{A})^2 \right) d\mathcal{A} \tag{14}$$

is the regularization term where $\tilde{C}$ and $\tilde{A}$ are priors, $\mathcal{A}$ is the mesh area and $\gamma_{sC}, \gamma_{aC}, \gamma_{sA}, \gamma_{aA}$ are regularization parameters that determine how much the solution is penalised in terms of smoothness and deviation from the priors and $I$ is the misfit term

$$I = \frac{1}{2\mathcal{A}} \int \left( [u - u_{obs}] / u_{err} \right) d\mathcal{A} + \frac{1}{2\mathcal{A}} \int \left( [v - v_{obs}] / v_{err} \right) d\mathcal{A} + \frac{1}{2\mathcal{A}} \int \left( \left[ \dot{h} - \dot{h}_{obs} \right] / \dot{h}_e \right) d\mathcal{A}, \qquad (15)$$

where $u_{obs}$, $v_{obs}$ and $\dot{h}_{obs}$ are observations of surface ice velocity and changes in ice thickness and $u_{err}$, $v_{err}$ and $\dot{h}_e$ are their respective observational errors.

By minimising the misfit to both velocities and thickness changes, rather than just velocity (as has generally been done in the past), we can incorporate additional information that helps add further constraints on the inverted $A$ and $C$ fields. However,
these two observational datasets are derived independently and generally not consistent with one another, meaning that our model will not be able to perfectly fit both the observed velocity and observed thickness change at the same time. A balance needs to be found such that our model matches both datasets as well as possible. To this end, we take the error terms as given for the $u_{err}$ and $v_{err}$ but use a spatially uniform $\dot{h}_e$ whose magnitude is one of our uncertain model parameters. The effect of this is that $\dot{h}_e$ becomes a scaling parameter that determines to what degree the inverse procedure tries to match observed
thickness changes as against observed velocities. Details of the observations and errors described above are found in Sec. 2.3.

### 2.1.5 Ice sheet geometry

To define our ice sheet geometry for the two sets of simulations, we require data for the bedrock elevation ($B$), surface elevation ($s$) and ice thickness ($h$). We assume that $B$ does not change throughout all our simulations, and here we use gridded data from BedMachine Antarctica version 3.4 (Morlighem, 2022). The Úa-*fwd* simulations, starting 2021, use the same dataset to define
$s$ and $h$. Measurements of these fields for the start of the Úa-*obs* simulations are more difficult to obtain and so rather than using these highly uncertain products, we use a similar methodology to De Rydt et al. (2021). This approach makes use of the precise measurements by satellite altimeter to work backwards from the present day geometry, subtracting the integrated surface elevation change trend over the period of interest, to calculate the ice sheet surface elevation, $s$, for 1996. This is fixed initially only for grounded ice that is further than 5 km from the coast in the present day, since altimeter measurements on
floating ice are less reliable. By assuming all floating ice is at hydrostatic equilibrium, ice thickness can be solved for given $s$ and $B$. Therefore, we make use of the position of the 1996 grounding line, for which we use the DinSAR derived grounding line of Rignot et al. (2014a), which approximately corresponds to the period 1992-1996. With this constraint on the extent of freely floating ice for our starting period, we iteratively adjust $s$ where it remains undefined, until the grounding line position of our new geometry matches that 1996 grounding line. We add a regularisation term that penalises spatial gradients in $s$, ensuring
that $s_{\text{bedmachine}} - s_{1996}$ does not vary by large amounts over small spatial scales and a smooth transition between the fixed $s$ field defined by altimetry and the $s$ field that we solve for.

### 2.1.6 Model Relaxation

Due to shocks in the transient solution immediately after initialization, at least partly caused by insufficient knowledge of bed geometry, all our model runs go through a brief period of relaxation of a few years. The model is initialized, using the inverse
methodology outlined in Section 2.1.4 to calculate $A$ and $C$, and then forced with constant initial surface and basal mass

balance for three years. The model is then re-initialized in an identical manner but with updated ice sheet geometry that arise from the relaxation period. Following re-initialization, the model is run for a further three year spinup with the same constant mass balance term, after which all model forcing terms evolve based on the CESM1 outputs for the period 1996-2021. A three year spinup was chosen as a balance between reducing large spurious thickness changes in the first few years of the simulation while also being as short as possible, and is consistent with Nias et al. (2016). Without this approach, widespread re-grounding occurs in shallow ice shelf cavities, and this approach greatly reduces this effect, although it persists somewhat in front of Pine Island glacier, often leading to some grounding line advance in this sector.

### 2.1.7   Adaptive remeshing

We make use of the Úa ice-flow model's adaptive remeshing capabilities, such that the finite element mesh evolves with the changing ice in the domain. For each set of simulations, an initial coarse resolution mesh is generated using mesh2d (Engwirda, 2014) and then immediately refined using nearest vertex bisection. Elements are refined in regions of high strain rate and close to grounding lines and calving fronts. By starting with a coarse initial mesh and refining using this approach, the mesh can both refine and un-refine as, for example, the grounding line migrates inland. Target mesh resolution along the grounding line is 400 m and along the calving front is 1500 m. Remeshing is done every year and with the same refinement criteria for all simulations. Our mesh refinement criteria generally lead to maximum and minimum mesh size during simulations of approximately 12.5 km and 400 m, respectively. The size of each triangle is defined as the leg of an isosceles right triangle having the same area as the triangular element. An alternative definition of the size of a triangular-shaped finite element is the length of the side of a perfect square with an area equal to that of the triangular element. Using this alternative definition, size estimates are a factor $\sqrt{2}$ smaller, and this smaller element size estimate is arguably more comparable to those used by finite-difference models.

### 2.2   Model forcing

Our ice sheet model is forced by changes at both the ocean and atmospheric boundaries. Motivated by the challenges in accurately representing ocean conditions, we drive our ocean melt parameterisation with the aid of recent state-of-the-art ocean model simulations of the region using MITgcm (Naughten et al., 2023). Rather than directly using the melt rates calculated in Naughten et al. (2023), which would not be representative of the state of our evolving ice shelf cavity geometry, we use modelled ocean conditions within the three main cavities to force our plume model parameterisation of basal melting. Specifically, we find the depth of winter water from the ocean model by searching for the coldest depth level in the top two-thirds of the water column, and extract temperature and salinity from this depth. Then, masking everything above the winter water core, the depth of the base of the thermocline is found by calculating $dT/dz$ for modelled temperature profiles and then searching for the first depth level (starting from the base) that exceeds a threshold value of $3 \times 10^{-3}$ (determined empirically). Temperature and salinity at this thermocline base depth are then extracted, meaning that for each ocean model profile we calculate a thermocline depth and two temperature and salinity values. These are spatially averaged over three predefined catchments (Pine Island, Thwaites and Dotson & Crosson) and sampled ten times per simulation year for each scenario and ensemble member. In our

Úa simulations we identify the catchment for each node and then find the closest corresponding point in time from which to sample these five scalar values. We take the temperature and salinity values and create vertical profiles that vary linearly from the thermocline depth to the surface values and are constant below the thermocline. Our catchments are defined everywhere in the ice flow model domain, so as to consistently apply the correct ocean conditions to newly floating nodes as the grounding line retreats. More details on the implementation of the plume parameterisation can be found in App. A.

The ocean model simulations described above are themselves forced with outputs from the CESM1 model (Kay et al., 2015; Sanderson et al., 2017, 2018). Results exist for 5 core scenarios representing different anthropogenic and natural forcing pathways. For our Úa-*obs* simulations we combine results from the *historical* scenario (1920-2005) with the RCP4.5 scenario (2005-2080) in order to span our desired period of 1996-2021. For this purpose, we select the RCP4.5 scenario that most closely matches observed atmospheric conditions in the model domain, by calculating a total RMSE in space and time compared to ERA5. For our Úa-*fwd* simulations we include two scenarios: RCP8.5 and Paris2C, which broadly represent more pessimistic or optimistic scenarios for anthropogenic forcing, respectively. These scenarios are further divided into 10 ensemble members each, which sample different realizations of internal climate variability within CESM1. Capturing this variability is important, since is likely to have a strong influence on the region and may have contributed considerably to observed trends (Holland et al., 2019, 2022). Therefore, all of our simulations randomly sample from these different ensemble members by sampling from a uniform distribution of ensemble IDs ($E_{ID}$), which are then one of the inputs to our surrogate model described in Sect.2.4.2 and thus contribute to our uncertainty estimates.

We use the same CESM1 atmospheric forcing as in Naughten et al. (2023) to provide changes in air temperature and precipitation during the course of our simulations. These two fields then drive the PDD component of our model (Sec.2.1.1), leading to changes in accumulation and surface melting. CESM1 modelled temperature and precipitation exhibit clear and systematic biases in the region which must be corrected for before they can be used to force the PDD model. Temperature shows an overall cold bias and differences in seasonal variability, which we correct for using processed data from 14 Automatic Weather Station (AWS) data in the region (Janet, Kohler Glacier, Noel, Kominko-Slade, Kathie, Ferrigno, Up Thwaites Glacier, AUstin, Toney Mountain, Lower Thwaites Glacier, Inman Nunatak, Bear Peninsula, Evans Knoll and Backer Island), obtained from the AntAWS Dataset (Wang et al., 2023). For precipitation bias correction, we compared CESM1 precipitation in the period 1998-2018 to precipitation from ERA5 reanalysis (Hersbach et al., 2017) and the MAR regional climate model (Marion et al., 2019; Donat-Magnin et al., 2020). In both cases, we create time-averaged but spatially varying interpolants from these data and calculate the difference to the time-averaged CESM1 field. This bias field is then added to the CESM1 forcing field in our model simulations and this same correction is done for all simulations. Note this is also the same methodology as in Naughten et al. (2023), meaning the atmospheric forcing in our ice sheet model is consistent with that of the ocean model.

### 2.2.1 Extended forcing

The model forcing described above, derived from climate and ocean modelling, only extends to the year 2100. To model ice sheet behaviour beyond this point in time, we require some method to extend our model forcing. Many different approaches could be used here, but our motivation in running these extended simulations stems from the inherently long response times

of the ice sheet to perturbations in climate, and so we aim to reveal a more complete picture of the ice sheet's response to the climate perturbations imposed in our main set of runs (2021-2100). Thus, in our extended simulations we detrend and then repeat the forcing from 2080-2100. For every point in the domain and for both temperature, precipitation and MITgcm forcing fields, a linear trend is calculated for the 2080-2100 period and then removed, and this new 20 year forcing is then applied every 20 years cyclically. By detrending the forcing, we keep climate relatively constant to allow the ice sheet more time to adjust to its new state, and furthermore we avoid large discontinuous jumps in the forcing terms after each 20 year repeat cycle. We prefer to extend our simulations with a cyclical forcing rather than driving the model with constant conditions in order to preserve potentially important natural variability (Jenkins et al., 2018).

## 2.3 Observations

Key to our uncertainty quantification approach is the use of observations to update our prior estimates of uncertain model parameters. We also require observations to initialise the ice-flow model for simulations starting in 1996 and 2021, as described in Sec. 2.1.4. For both purposes, we use satellite observations of surface ice velocity and surface elevation change over the entire ASE region. While satellite products have the significant advantage of providing unparalleled spatial coverage, ascribing a precise date to a particular field is often difficult, since velocity and surface elevation change maps are mosaics made up of many repeat satellite passes. Based on the variable and sometimes not well defined dates of the remote sensing products we use in this study, we have selected simulation start dates of 1996 and 2021 as the dates that align most closely with all these inputs taken as a whole.

Velocity observations for Úa-*obs* (i.e. initialised for 1996) are from the MEaSUREs InSAR data for the Amundsen Sea Embayment (Rignot et al., 2014b). These data originate from the ERS-1 and are dated from the 1st January to the 31st December 1996. Details of the error sources and estimation are provided in Mouginot et al. (2012), and the resulting term is the square root of the sum of the independent errors squared. Velocity observations for Úa-*fwd* are from the MEaSUREs annual Antarctic ice velocity map (Mouginot et al., 2017a). These observations are dated from 1st July 2019 to the 30th June 2020 and details of the errors can be found in Mouginot et al. (2017b). In addition to their use in model initialisation step described in Sect. 2.1.4, the two velocity maps described above are converted to surface ice speed and then differenced to provide one component of our observations with which we calibrate the forward model (Sec. 2.4).

Satellite derived $\dot{h}_{obs}$ are obtained from ITS_LIVE, which combines a number of satellite sensor observations from 17th April 1985 to 16th December 2020 (Nilsson et al., 2022; Nilsson and Paolo., 2023). These data are provided as a change in ice sheet elevation relative to December 16 2013, at a monthly resolution. For initialisation of each forward model we use an average of this field in the three years preceding the simulation start date, whereas for model calibration we calculate the difference in ice sheet elevation between 1st January 1996 and 16th December 2020. As described in Sec. 2.1.4, the $\dot{h}_e$ error term is treated as an uncertain model parameter for the purposes of model initialisation. For model calibration, we use the supplied error term, which is a root mean squared error as described in Nilsson et al. (2022).

## 2.4 Model calibration

The set of Úa-*obs* simulations are used to calibrate our model parameters, assigning probability distributions to these parameters based on how well they fit our observations of the region during the simulation period (1996-2021, Sec. 2.3). Model parameters contributing to uncertainty in our final estimates of SLR contribution are listed in Table 1. Nine of these parameters are included in the Bayesian inference step outlined in Sec. 2.4.3. Probabilities for the remaining three parameters, related to SMB ($p$ and $\sigma_T$) and calving ($C_Q$), are specified separately due to there being insufficient model sensitivity to these parameters in the present day or over the relatively short observational period, as described in App. C.

### 2.4.1 Dimensionality reduction

Approximating probability distributions for large numbers of parameters and a complex model may require evaluating the forward model sequentially hundreds of thousands of times and the computational cost of our forward model makes this intractable. As an alternative, we use the Úa-*obs* simulations as the training set for a surrogate model that is orders of magnitude faster to run than the forward model and therefore allows proper sampling of the probability distributions. Therefore, we require a surrogate model that takes as input a set of model parameters $\theta$ and outputs an expected change in ice speed and surface elevation that we can compare to observations.

Our set of training simulations suffers from the curse of dimensionality, since any model output we are seeking to emulate is defined at every node in the domain, but the information within these simulations is relatively sparse. We use singular value decomposition (SVD) to decompose our set of training simulations into principal components that make use of the strong spatial correlations in the model results. We build an $m \times n$ matrix $Y$ of Úa-*obs* simulation results, where each row is an individual model run $y_i$ in the $m$ member ensemble for a particular combination of model parameters $\theta_i$ from the full sample of model parameters $\Theta$, i.e. $y_i = \mathcal{F}(\theta_i)$, each column represents a particular node and $y$ is either the modelled surface elevation change or surface speed change during the course of the simulation. Since the finite element mesh of each simulation evolves dynamically depending on the transient state of that simulation, the column entries of matrix $Y$ arise from mapping model results to a single finite element mesh, consisting of $n$ nodes, using the underlying shape functions. SVD involves finding the matrices $U$, $S$ and $V$ such that

$$Y = USV^T, \tag{16}$$

where the columns of $U$ and $V$ are the left and right singular vectors and they are both orthogonal such that $U^T U = V^T V = I$. $S$ is a diagonal matrix where the diagonal entries are the singular values of $Y$ and are ordered from largest to smallest. We define the matrix $B = US$ which represents the new basis and each row of $B$ represents the main modes of change between model simulations. The first four rows of $B$ are plotted for $Y_{sec}$ and $Y_{vel}$ in Figure 2, showing that the largest principal component captures an overall thinning and retreat signal for the entire ASE while subsequent principal components represent variation in this response between the main catchments. By using only the first $k$ columns of these matrices, we end up with a truncated

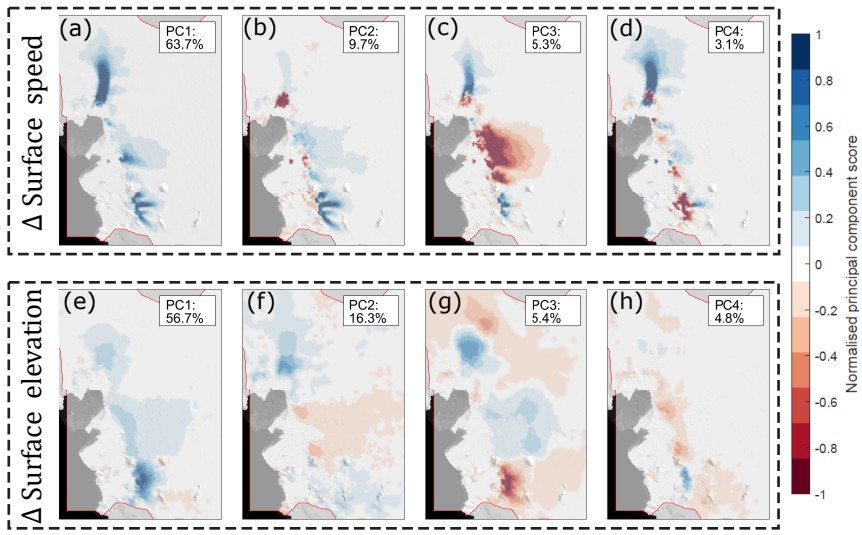

**Figure 2.** First four principal components for changes in surface ice speed (top row) and surface elevation (bottom row). The percent of the total variance captured by each component is given in the top right corner of each panel. The shadded region surrounded with a red border indicates the extent of the model domain. Background image is from the MODIS mosaic of Antarctica 2008-2009 (Haran, 2014).

SVD. The proportion of total variance captured for a given integer $k$ is

$$f(k) = 1 - \frac{\sum_{i=1}^{k} S_{ii}}{\sum_{i=1}^{m} S_{ii}} \tag{17}$$

and for each of the two $Y$ matrices we choose $k$ such that $f(k) > K$, where $K$ is a hyperparameter as described in Sec.2.4.2, and our final surrogate used $k = 8$ for surface elevation change and $k = 11$ for change in surface ice speed. This results in an approximation for $Y$, given by $\tilde{Y} = \tilde{B}\tilde{V}^T$, where $\tilde{B}$ and $\tilde{V}$ contain the first $k$ rows of $B$ and $V$.

In order to conduct our Bayesian inference in this newly defined framework, we need a way to project fields defined on our finite element mesh into the reduced $k$ dimensional space. Premultipling by $\tilde{B}^T$, followed by $(\tilde{B}^T\tilde{B})^{-1}$ yields the desired mapping from $n$-dimensional field $y$ to the $k$-dimensional field $\hat{y}$

$$(\tilde{B}^T\tilde{B})^{-1}\tilde{B}^T y = \hat{y}. \tag{18}$$

### 2.4.2 Surrogate model

The SVD methodology outlined above provides a way to drastically reduce the dimensionality of our problem by encapsulating each model simulation as a linear combination of $\tilde{B}$ and $\tilde{V}^T$. The matrix $\tilde{B}$ represents the main modes of change as exhibited in the set of training simulations in $Y$, and each row of $\tilde{V}^T$ represents the proportion of each of these $k$ components for a particular set of model parameters. Since all our Bayesian inference is done in principal component space, our surrogate model

is trained so that, for a particular combination of $\theta$, it can accurately predict the corresponding row of $\tilde{V}^T$ for both speed and surface elevation change (simultaneously). Outside of the existing simulations in Úa-*obs*, we do not know this mapping a-priori. Thus, the input data used to train our surrogate model consists of the matrix of sampled model parameters $\Theta$ and the network targets are the corresponding rows of $\tilde{V}^T$. Following a similar approach to Brinkerhoff et al. (2021), we train a deep residual neural network for this purpose. A detailed description of the design and training of this surrogate model can be found in App. D. Once trained, the computationally efficient surrogate model can be used in combination with our parameter priors for Bayesian inference, as described in Sec. 2.4.3.

Before moving forwards with the Bayesian inference, we must verify that the truncated SVD and the surrogate model (reprojected back to model space using the SVD) are able to properly represent the observations. It is entirely possible, if there are no model simulations in $Y$ that show similarities to the observations, that there does not exist a $k$-dimensional $\hat{y}$ that matches closely to the observations. Similarly, although the surrogate model is trained to agree well with the forward model across the parameter space, there is no guarantee that the output of the model for the optimal set of model parameters agrees well with observations. To verify these two points, in Fig. 3 we compare the observations of change in surface elevation and surface ice speed with their equivalents, having been reprojected using the truncated SVD, and with the surrogate model output for the maximum a posteriori parameters (as described further below). This confirms that both the magnitude and spatial distribution of thinning and acceleration in the region are well approximated using the truncated SVD and our surrogate model.

### 2.4.3 Bayesian inference

We assume that within the complete parameter space $\Theta^*$, there exists a combination of parameters $\theta'$ that leads to a good agreement between model output $\mathcal{G}(\theta')$ and our chosen set of observations in the basis representation $\hat{y}$ (note that $\Theta \subset \Theta^*$ and presumably $\theta' \nsubseteq \Theta$). Our search for $\theta'$ and its associated uncertainty is complicated by the presence of noise in the observational data and errors in our forward model. We use a Bayesian framework to infer probability distributions for our model parameters by updating our prior beliefs with observations to obtain a posterior probability. Using Bayes theorem, we can express this posterior probability of certain model parameters given the observations as

$$\pi(\theta \mid y) = \pi(\theta) \cdot \mathcal{L}(y \mid \theta) \tag{19}$$

where $\pi(\theta)$ is the prior probability for $\theta$ and $\mathcal{L}$ is the likelihood function. The choice of priors is an important step in any Bayesian analysis, and a more detailed description is given in Appendix B. For model parameters that do not have a directly measurable impact within our model, we choose uniform priors bounded by upper and lower limits chosen to span physically plausible values (see App. B). For parameters related to the surface and basal mass balance models which calculate measurable quantities independently from the ice flow model we use priors constrained on those observations.

The output of our surrogate model $\mathcal{G}$ is related to the observed changes in the basis representation by

$$\hat{\mathbf{y}} + \mathbf{e}_{obs} = \mathcal{G}(\theta) + \mathbf{e}_{model} \tag{20}$$

where $\mathbf{e}_{obs}$ and $\mathbf{e}_{model}$ are the measurement and model discrepancy terms, respectively. Furthermore, the error of our forward model can be broken down into two terms, i.e. $\mathbf{e}_{model} = \mathbf{e}_{Ua} + \mathbf{e}_{rnn}$, which are the errors in the forward ice sheet model and

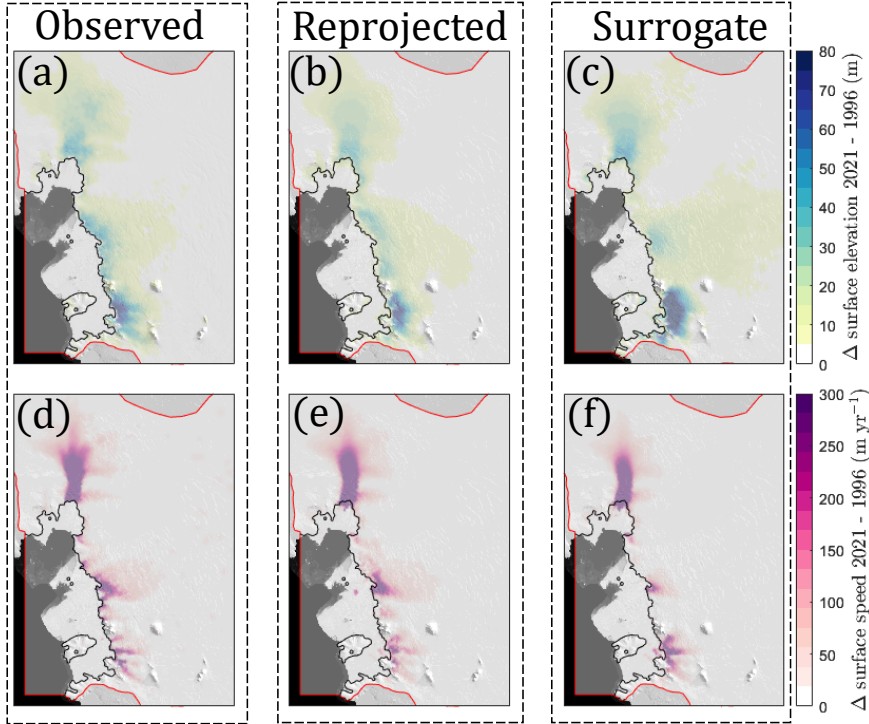

**Figure 3.** Comparison between observed (left column), reprojected via truncated SVD (middle column) and surrogate (right column) changes in surface elevation (top row) and surface ice speed (bottom row) for the period 1996 to 2021. Positive values indicate thinning and acceleration. Background image is from the MODIS mosaic of Antarctica 2008-2009 (Haran, 2014).

the discrepancy between the forward model and the surrogate model, respectively. We assume that the three discrepancy terms ($\mathbf{e}_{obs}$, $\mathbf{e}_{Ua}$ and $\mathbf{e}_{rnn}$) are statistically independent and Gaussian distributed and we require an expression for the covariance matrix for each of these terms.

Observational errors are assumed to be uncorrelated, i.e. $\mathbf{e}_{obs} \sim \mathcal{N}(0, \boldsymbol{\sigma}_{obs}^2 \mathbf{I})$, where $\mathbf{I}$ is the identity matrix. Observations of surface elevation change and speed are derived from satellite measurements and processing of these data includes (spatially varying) estimates of uncertainty. These supplied error estimates are derived from various sources including instrumental accuracy and the number of data points in a particular grid cell.

We can generally expect errors in the ice sheet model to be spatially correlated, since if the model is not able to accurately capture ice flow at a given node it is likely that the neighbouring nodes will also be similarly inaccurate. Thus, the discrepancy covariance will not be diagonal and so $\mathbf{e}_{Ua} \sim \mathcal{N}(0, \boldsymbol{\Sigma}_{Ua})$. We use an exponential covariance function to represent the spatial correlation in $\boldsymbol{\Sigma}_{Ua}$, i.e.

$$\boldsymbol{\Sigma}_{Ua} = \sigma_{Ua}^2 \exp\left(-\frac{\|x_i - x_j\|^2}{2\ell^2}\right) \tag{21}$$

where $\ell$ is a length scale and $\sigma_{Ua}^2$ is the model discrepancy variance. This variance term, representing how well the model is able to match observations, is hard to quantify but one reasonable argument is that we are not able to model the ice sheet more accurately than we can observe it, i.e. $\sigma_{Ua}^2 > \boldsymbol{\sigma}_{obs}^2$. Here, we conservatively set the model discrepancy variance to be four times the mean observational error, i.e. $200 \, \mathrm{m \, yr^{-1}}$ in surface ice speed and $20 \, \mathrm{m \, yr^{-1}}$ in $dh/dt$. We can estimate the model error correlation length scale $\ell$ by fitting an exponential semi variogram to the model results, whereby the range of the semi variogram is equivalent to $\ell$ in Eq.21.

Finally, the discrepancy between the ice sheet model and its surrogate can be estimated directly using the test set, hidden from the network during training. In this case, since the network is making a prediction in the reduced $k$-dimensional space and each component is orthogonal, we can expect these to be spatially uncorrelated and thus $\mathbf{e}_{rnn} \sim \mathcal{N}(0, \boldsymbol{\sigma}_{rnn}^2 \mathbf{I})$. With these assumptions,

$$\boldsymbol{\sigma_{rnn}^2} = \frac{\sum_{i=1}^{n_{test}} \left( \mathcal{F}(\theta_i) - \mathcal{G}(\theta_i) \right)^2}{n_{test} - 1} \tag{22}$$

where $n_{test}$ is the number of simulations in the test set (566).

Importantly, the ice sheet model and observation discrepancy terms defined above are defined in model space, whereas the surrogate model discrepancy is defined in the reduced $k$-dimensional space. Thus the final total covariance matrix, defined in the reduced $k$-dimensional space, is given by

$$\hat{\boldsymbol{\Sigma}} = A \left( \sigma_{\mathbf{obs}}^2 \mathbf{I} + \boldsymbol{\Sigma_{Ua}} \right) A^T + \sigma_{rnn}^2 \mathbf{I} \tag{23}$$

where $A = (\tilde{B}^T \tilde{B})^{-1} \tilde{B}^T$ follows from Eq. 18. With this estimate of the covariance matrix, we can calculate the likelihood term in Eq. 19, which is given by

$$\mathcal{L} = \prod_{i=1}^{N} \frac{1}{\sqrt{(2\pi)^{N_{out}} \det(\hat{\boldsymbol{\Sigma}})}} \exp\left( -\frac{1}{2} (\hat{y}_i - \mathcal{G}(\theta))^T \hat{\boldsymbol{\Sigma}}^{-1} (\hat{y}_i - \mathcal{G}(\theta)) \right). \tag{24}$$

In practice, the posterior probability given in Eq.19 does not have a tractable closed-form solution for such a complex model, so we use a Markov-Chain Monte Carlo (MCMC) algorithm to approximate its form numerically. For this we use UQLab (Marelli and Sudret, 2014), which provides an extensive suite of uncertainty quantification tools for use within MAT-LAB. MCMC algorithms construct a Markov chain that explores the parameter space, whereby a 'chain' is a particular set of parameters in a given iteration. In each iteration, a new set of parameters is proposed in each chain and these are either accepted or rejected. After a sufficient number of iterations, the Markov chain converges such that the distribution of samples closely approximates the true posterior distribution. We used the Affine Invariant Ensemble (AIES) algorithm to determine whether a proposed step for each chain should be accepted, due to its strength in dealing with correlation between parameters and only requiring one tuneable parameter ($a$, which we set to 1.5). Further details on the AIES algorithm and the choice of $a$ can be found in Goodman and Weare (2010); Allison and Dunkley (2013). We ran the AIES algorithm with 20 chains for 5000 iterations and verified convergence using the Gelman-Rubin diagnostic (Gelman and Rubin, 1992).

Once the Markov chains have converged, we take the final 1000 iterations for each chain and infer the marginal distributions and copula, once again using the uncertainty quantification tools within UQLab. Histograms show the converged samples for

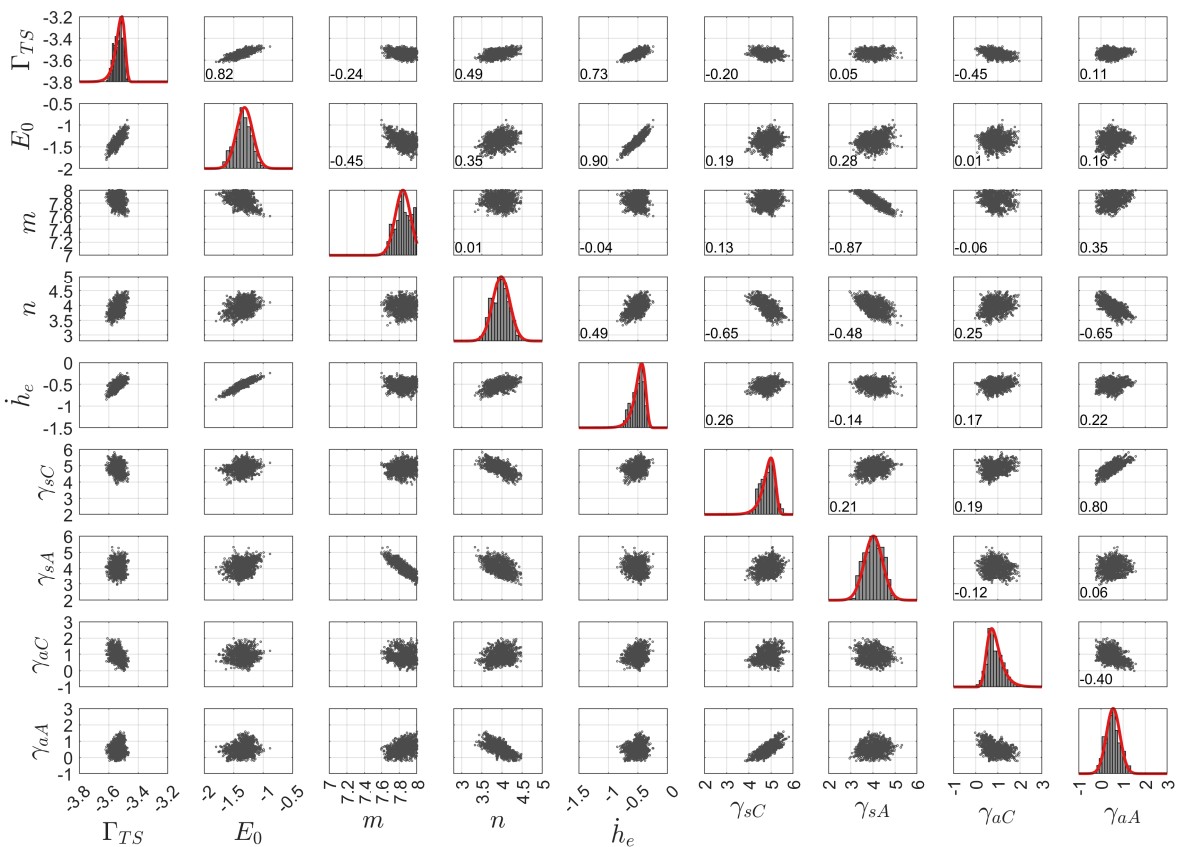

**Figure 4.** Parameter distributions for each converged chain in the final 1000 iterations of the MCMC algorithm. Red lines in the panels of the main diagonal show the fitted probability distribution for each parameter, blue lines show the priors (not shown if a uniform prior was used). The number in the bottom left corner of the upper diagonal plots shows the correlation coefficient between each two sets of parameters.

each chain, along with the fitted marginals are plotted along the main diagonal of Fig. 4. The off-diagonal elements show the joint probability distribution for each pair of model parameters, helping to reveal correlations between certain model parameters (correlation coefficient between the pairs of parameters is included in the bottom left corner of the upper diagonal panels). An interesting side note here is that our probabilistic approach finds a distribution for $n$ centered around 4. This lines up well with other recent studies which have used different approaches to suggest that a value closer to 4 than 3 may be more appropriate (Qi and Goldsby, 2021; Millstein et al., 2022). Having inferred the probability distributions for each parameter, we use latin hypercube sampling to generate our sample of model parameters, with which we can run our forward model.

## 3 Sea level rise projections

We run our forward model until 2100, sampling from parameter probability distributions as calculated in Sec. 2.4. Sea level contribution is calculated as change in ice volume above flotation, assuming an area of the ocean of $3.625 \times 10^{14}$ m$^2$, meaning that 1 mm of sea level contribution is equivalent to $\sim$362.5 Gt of water equivalent ice added to the ocean. Once again, our uncertainty quantification approach requires a large number of forward model evaluations, and so we train a deep surrogate model that takes as input a vector of model parameters along with the forcing ensemble ID ($E_{ID}$) and outputs cumulative change in ice sheet volume above flotation in each modelled year from 2021 to 2100. This is a different surrogate to that described in Sec. 2.4.2 and details of the network architecture, its training and validation can be found in App. E. The surrogate is trained separately to make predictions for the two emissions scenarios, from a total training set of 2290 simulations, of which 20% are held back for validation and testing. The RMSE between forward model (target) and surrogate (predicted) cumulative change in volume above flotation by the year 2100 was 1.65 mm and 1.35 mm for the RCP8.5 and Paris2C scenarios, respectively. Expressed as a percentage error between targets and predictions, the surrogate model error was 5.7% and 9.5% for Paris2C and RCP8.5, respectively.

Using the surrogate model we calculate a timeseries of sea level contribution until 2100 for one million samples drawn from the posterior parameter probability distributions, allowing for dense sampling of the parameter space and robust estimation of the probability distribution for each year. We show the resulting sea level curves for the RCP85 and Paris 2C scenarios, along with selected confidence intervals, in Fig. 5. This shows that there is almost no difference in sea level contribution by the year 2100 for the two scenarios and the uncertainties are almost identical, although RCP8.5 has a slightly wider range of extremes than Paris 2C. Results from a one million member ensemble of the Paris2C scenario show a median SLR contribution of 19.0$\pm$2.2 mm from the ASE by 2100, and 18.9$\pm$2.7 mm for RCP8.5. The 5-95% percentiles (the so-called "very likely range" in IPCC reports) are 15.6-22.9 mm and 15.1-24.0 mm for Paris2C and RCP8.5. These projections represent a similar but generally slightly lower estimate than most previous estimates, as discussed in more detail in Sec. 6.

To explore the response of each glacier in more detail we look to the Úa-*fwd* simulations directly, rather than the surrogate model that only provides information on the sea level contribution. In Fig. 6 we show the final grounding line and calving front positions for all simulations in the year 2100, compared to their starting positions in 2021. To generate this plot, the domain is divided into cells and the colormap represents how frequently a grounding line (brown) or calving front (green) lies in a given cell as a percentage of all grounding lines or calving fronts, i.e. 10% would mean that a grounding line or calving front was present in this cell for 10% of the total number of simulations. This figure shows extensive calving front retreat across the entire ASE, with an almost complete collapse of the Dotson and Crosson ice shelves and only very small ice shelves remaining in front of Pine Island and Thwaites glaciers. Grounding line movement in the region is generally more limited, with very few model simulations finding either significant re-advance or retreat by 2100. Some limited re-advance occurs frequently at the grounding line of Pine Island Glacier. This is most likely as a result of inaccurate bed topography rather rather than a physical response of the glacier to climatic forcing, and this may lead to a slightly negative bias in terms of SLR contributions from this glacier.

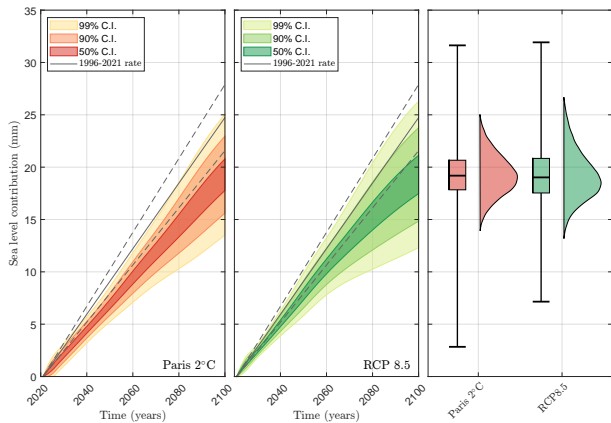

**Figure 5.** Sea level contribution from the ASE for the Paris 2C and RCP8.5 emissions scenarios, as calculated by the surrogate model. Also plotted is the mean observed 1996-2021 rate for the region (solid black line) and its propagated uncertainty (dashed black lines) as calculated using the input-output method (Davison et al., 2023). Shading indicates the 50, 90 and 99% confidence intervals. Box plots in the right panel show the mean, interquartile range and full range of SLR contributions for each emissions scenario.

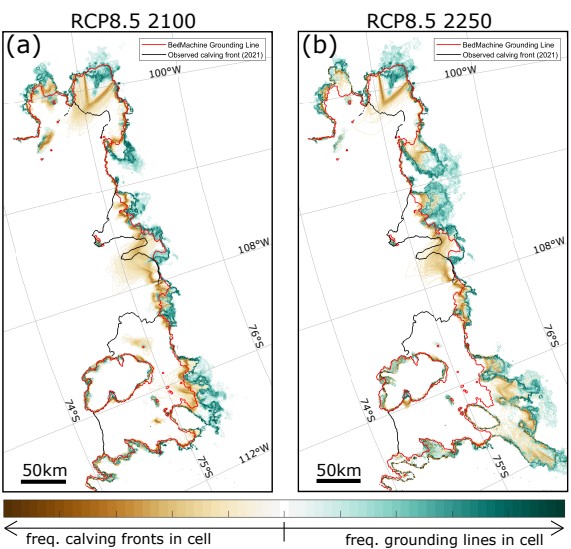

**Figure 6.** Grounding line and calving front positions for all Úa-*fwd* simulations (panel a) and Úa-*extended* simulations (panel b) for the RCP8.5 scenario. Locations of the grounding line (red line) and calving front (black line) in 2021 are also shown. Model results are transferred to a uniform grid and the colormap indicates the proportion of simulations for which a grounding line (blue-green colormap) or calving front (brown colormap) is present in a particular grid cell.

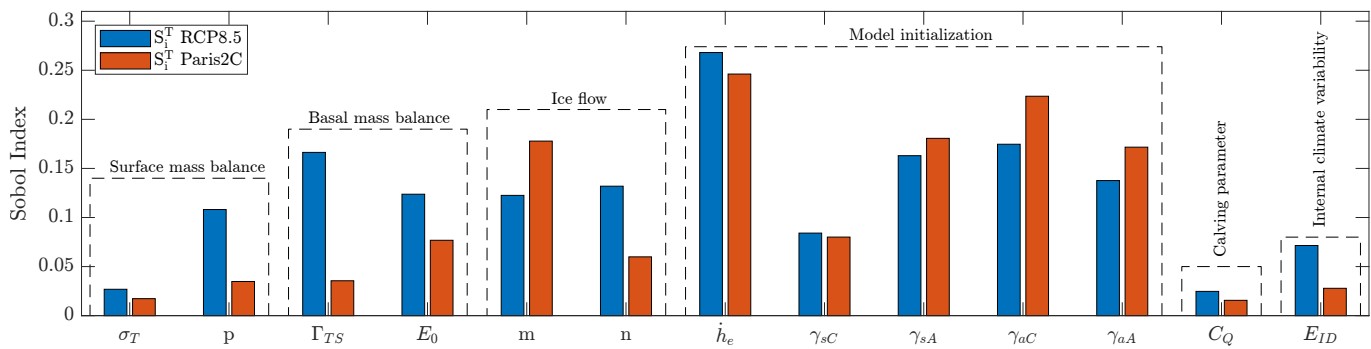

**Figure 7.** Sobol indices for each model input, representing the fractional contribution of each input on the uncertainty in our projections of sea level contribution for the ASE. Each input consists of two columns, showing the Sobol index as calculated for either the RCP8.5 or Paris 2C scenarios.

## 4   Sources of uncertainty

We can explore the relative contribution to total uncertainty arising from different model parameters and internal climate variability using Sobol indices. These are based on the principle that the forward model can be expanded into summands of increasing dimension, such that the total variance is then given by the sum of the variance of these summands. The first-order Sobol indices ($\mathcal{S}_i$) represent the effect that only parameter $\theta_i$ has on the variability of the model response ($Y$), i.e.

$$\mathcal{S}_i = \frac{\text{Var}\left[\mathbb{E}(Y|\theta_i)\right]}{\text{Var}(Y)} \tag{25}$$

Higher order Sobol indices give the interactions between parameters and the total Sobol index for a parameter is then the sum of all Sobol indices. We plot the total Sobol indices for each model input in Fig. 7. The first twelve Sobol indices for each scenario represent parametric uncertainty while $E_{ID}$ represents uncertainty resulting from internal climate variability.

Overall, the $\dot{h}_e$ parameter related to our inversion is the single most important for both scenarios, and all inversion parameters together contribute 51% and 66% of total uncertainty for RCP8.5 and Paris2C, respectively. Parameters $m$ and $n$, related to ice flow, contribute 17% and 19% while parameters related to basal melting ($\Gamma_{TS}$ and $E_0$) contribute 18% and 8% for RCP8.5 and Paris2C respectively. Generally, Sobol indices for the RCP8.5 experiments show a stronger sensitivity to parameters related to external forcing (e.g. mass balance and internal climate variability) whereas the Paris2C simulations are more sensitive to parameters related to internal ice dynamics and initialisation. This makes sense since the clearest difference between the two scenarios is changes in mass balance and particularly an increase in precipitation in the RCP8.5 scenario.

The most notable finding in terms of Sobol indices is the strong sensitivity of our sea level projections on parameters related to model initialization. It implies a strong sensitivity to the resulting $A$ and $C$ fields whose spatial distribution these parameters alter. This in itself is not surprising, since uniform $A$ and $C$ fields would yield model simulations that have no bearing on the present state of the ice sheet and would have no meaningful predictive power. Additionally, our Sobol indices are calculated for predictions spanning a 79 year period, i.e. a relatively short term 'forecast' in terms of ice sheet modelling. We would expect

that for longer simulations spanning hundreds of years this sensitivity to initialisation would become weaker, although since the physical interpretation behind $A$ and $C$ are processes such as hydrology that generally vary with time then using fields that are constant in time becomes harder to justify over longer timescales.

It is important to emphasise that with this methodology we can only explore certain sources of uncertainty, mostly related to model parameters. Given the important role that surface mass balance appears to play in our simulations to in offsetting increased grounding line flux, resulting in sea level projections at the lower end of published results (Sect. 5.1), it is clear that atmospheric forcing is an important consideration. We can explore uncertainty related to our PDD model but not the precipitation and temperature forcing from CESM1 that drives this. Clearly an important avenue for future research should be to include forcing from other models and indeed the structural uncertainty associated with our choice of ice sheet and ocean model.

## 5 Extended simulations

We conduct two sets of simulations that extend a subset of the Úa-*fwd* simulations from their end date of 2100 to either 2250 or 2300. The first set of these extended simulations, to the year 2250, continues a large subset (608) of Úa-*fwd* model runs with the goal of exploring how the ASE sea level contribution evolves over a longer period of time in response to the changes in forcing imposed until 2100. The second set, extended to the year 2300 and saving all model fields at the end of each year, was conducted to enable a more detailed analysis of our simulations and was necessarily much smaller (60 simulations) due to computational constraints. Note that since our extended ocean forcing is detrended and repeated from 2100, SLR estimates from these extended simulations may be conservative compared to studies where forcing continues to evolve with time.

### 5.1 Extended simulations to 2250

We randomly selected a total of 608 simulations from the Úa-*fwd* model runs and continued them from where they stopped in 2100, using the extended forcing described in Sect. 2.2.1. We show the cumulative sea level contribution for the ASE for the RCP8.5 and Paris 2C emissions scenarios in Fig. 8a. By 2250 there is a more substantial difference between the two scenarios, with a median sea level contribution of 6.0 cm and 7.1 cm for RCP 8.5 and Paris 2C, respectively. These results are similar to recent modelling work using a calibrated melt rate parameterisation that found a maximum of 8 cm of sea level contribution by 2300 (Reese et al., 2023), but substantially lower than the ~0.3 m by 2200 in response to a +3°C global warming scenario (DeConto et al., 2021).

Interestingly, our results show a generally greater sea level contribution under the Paris2C emissions scenario. To explore the reason for this in more detail, we compare the area integrated precipitation, total grounding line flux and change in ice volume above flotation in the final year of these extended simulations. This shows very little difference between scenarios in terms of dynamic ice loss, whereas there is a relatively large difference in terms of total precipitation. These results suggest that over longer timescales increases in precipitation for RCP8.5 may outweigh the relatively minor differences in ocean forcing,

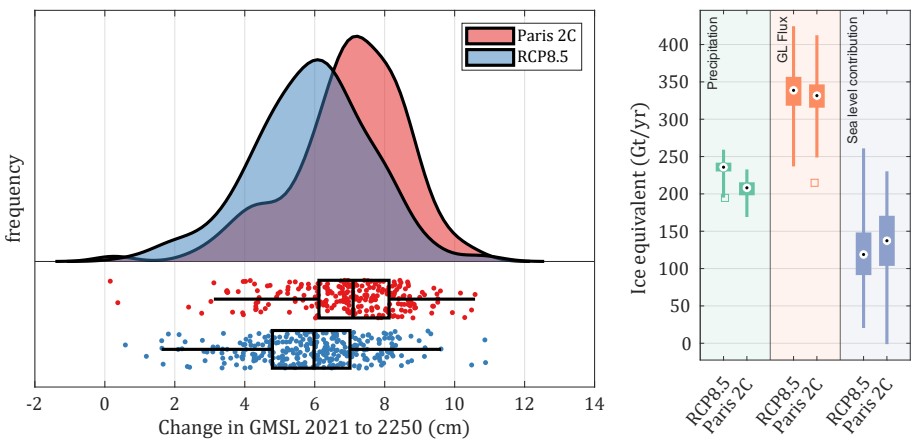

**Figure 8.** Extended simulation results, showing change in global mean sea level from the ASE region between 2021 and 2250 for the RCP8.5 and Paris 2C emissions scenarios (panel a) and the relative contribution of different ice mass terms for the final year of simulations (panel b).

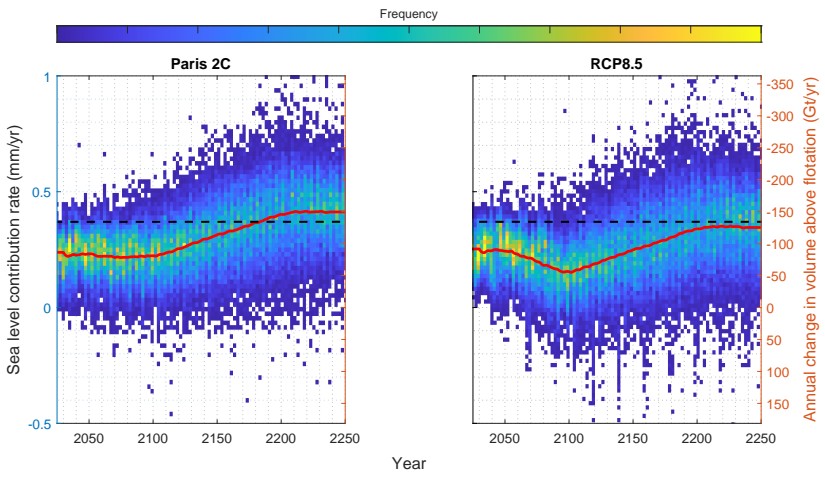

**Figure 9.** Annual rate of sea level contribution from the Paris2C and RCP8.5 scenarios for all extended simulations. The red line shows the average across all simulations for each scenario, the dashed black line shows the observed rate from 1996 to 2021 (Davison et al., 2023) and the background colormap intensity represents how many simulations fall within a particular year/sea level bin.

relative to the Paris 2C scenario, although with the caveat that our ocean forcing does not evolve beyond 2100 and may itself show a stronger scenario dependency over longer timescales.

To better understand how the regional mass loss evolves during the course of our simulations, we plot the rate of volume above flotation loss (or equivalently sea level contribution rate) for all simulations and both scenarios (Fig. 9). In both scenarios, there is a clear increase in the rate of ice loss after 2100, with the rate approximately doubling from 2100 to 2250 in both cases.

In addition, a clear difference emerges between the Paris2C and RCP8.5 scenarios up to approximately the year 2100. RCP8.5

shows a substantial decrease in the rate of mass loss that is not present in the Paris2C scenario, but following this both scenarios follow very similar trajectories. By the year 2250, average annual rates of SLR contribution from the ASE reach 0.34 and 0.41 mm yr$^{-1}$ for RCP8.5 and Paris 2C, respectively. This represents a similar rate of SLR to the observed rate of 0.37 mm yr$^{-1}$ observed in the period 1996-2021 (Davison et al., 2023). The majority of our model simulations start with a slightly lower SLR contribution rate than the observed rate (dashed black line in Fig 9). This is in spite of the fact that our model is calibrated based on observed thinning and acceleration and one of the terms that the inverse initialisation seeks to minimise is the the observed thinning rate $\dot{h}_{obs}$. However, as explained in Sec. 2.1.4, simultaneously matching both observed thinning rate and observed velocities was not possible, given the ice geometry from BedMachine. Trying to minimise both, in combination with our relaxation step, generally leads to an initial grounding line flux that is less than observed estimates, however both these steps lead to greatly improved model behaviour over longer timescales (see also Sec. 2.1.6).

## 5.2 Extended simulations to 2300

From the set of simulations described in Sect 5.1, we evenly sampled 30 simulations from the sea level contribution distribution for both the Paris2C and RCP8.5 forcing scenarios, resulting in a total of 60 additional simulations extended to the year 2300. With such a small subset of simulations we do not capture the full uncertainty in our simulations, but this sample allows us to analyse simulations in much more detail, to extract information that can help understand our model behaviour. In Fig. 10 we plot a timeseries of important metrics for each of the main catchments in our model domain.

In terms of overall change during these extended simulations, the Dotson and Crosson catchment shows the largest reduction in both grounded and floating area, with a corresponding increase in grounding line flux. Total ocean-induced melt follows the same trajectory as ice shelf area, with a very significant decrease until 2100, at which point most of Dotson and Crosson ice shelves are completely gone and melt is forcibly limited to relatively small ice shelves in front of the major outlet glaciers (see also Fig A2). This catchment also shows a decreasing trend in accumulation for both scenarios and an initial increase in surface melting only lasts until 2100 before reducing again, most likely as a result of the loss of floating shelf area where most of the melting takes place.

Turning our attention to Pine Island glacier, we find an initial increase in grounding line flux generally levels out by 2100, at which point the two scenarios diverge slightly with an average reduction in flux for Paris2C. Although the average grounding line flux timeseries appears relatively smooth, looking more closely at individual simulations we can see these are punctuated by periods of greatly enhanced flux, whose timing and magnitude are likely a result of periods of grounding line retreat. The clearest overall signal in terms of mass balance is in the accumulation term, which initially increases until 2100, particularly for RCP8.5, but then levels out as a result of our extended forcing approach (see Sect. 2.2.1). Ocean melting for both scenarios decreases by $\sim 20$ Gt yr$^{-1}$ by 2100 as the floating area is reduced, at which point it levels off as the increasing grounding line depth offsets any continued loss in ice shelf.

Thwaites glacier shows only relatively minor changes in grounding line flux, despite considerable grounding line retreat (see also Fig. 6). Thwaites Eastern ice shelf almost completely disappears in most simulations but this loss in floating area is offset by the regrowth of the western ice shelf, promoting a large increase in ocean melt rates for this area. Accumulation shows a

590 similar trend to Pine Island glacier, with an increase until 2100 that is stronger in RCP8.5 than Paris2C scenario. Arguably the most noteworthy feature in our simulations is that generally Thwaites glacier does not appear to show strong acceleration in response to either grounding line retreat or loss of floating ice shelves. Grounding line retreat continues steadily until 2300, so this is not the result of the grounding line becoming stuck on some portion of the bed. One possible explanation is the formation of a new branch of Thwaites glacier in almost all simulations, flowing out into Northwest Pine Island Bay (Fig. 6).

As the grounding line retreats further and this new branch grows, it redirects much of the previous Thwaites glacier flow and accounts for almost 50% of the total grounding line flux for this catchment. The key difference to the present day situation is the emergence of a buttressed ice shelf in front of this outlet, counteracting the increased mass loss that might be expected from such a retreated configuration.

Overall, our simulations show a complex response that varies considerably across each of the three regions. Taken together,
these result in little change or even a decrease in the rate of SLR contribution until 2100 (Fig. 9), at which point mass loss accelerates, reaching or exceeding present day observed rates by 2200. Mass loss due to ocean melting decreases substantially by 2100, from $\sim 300 \mathrm{Gt\,yr^{-1}}$ to $\sim 200 \mathrm{Gt\,yr^{-1}}$, averaged across simulations and scenarios, due mostly to a large reduction in floating area during this time. Conversely, accumulation increases over the same period. The acceleration in SLR contribution after 2100 is largely due to increased grounding line flux in the Dotson and Crosson region, together with a cessation of the
increasing surface accumulation as we remove trends in the atmospheric forcing after 2100. Increased rates of SLR contribution persist until 2200, at which point they stagnate once again as grounding line flux from the Dotson and Cross region decreases.

A crucial factor to consider when investigating future sea level contribution of the ASE is whether or not a tipping point may be crossed, in which a positive feedback yields a greatly increased rate of ice loss. Robustly identifying tipping points with this set of simulations is not possible since we would require reversibility experiments or other tipping point analysis
methods (e.g. Rosier et al., 2021), however looking at results from these longer simulations may provide some indirect insight. Crossing a major marine ice sheet instability type tipping point would be expected to manifest itself in a marked increase in grounding line flux that stands out from other experiments, and we do not see any behaviour of this kind in any of our simulations. Similarly, a marine ice cliff instability type tipping point should lead to a very large increase in calving flux and although some simulations show quite a large three to fourfold increase in the Dotson and Crosson region, these increases
do not appear to be self-reinforcing and leading to a runaway retreat. An important caveat is that, since the ocean forcing is held quasi-constant after 2100, our interpretation on tipping points in these extended simulations is limited to possibly delayed response to perturbations in forcing up to 2100. If we were to force our model with a continued increase in ocean thermal forcing, it may be that a tipping point could be crossed. In the ISMIP6-2300 experiments, for example, where ocean thermal forcing increases considerably after 2100 (Fig. A1), some simulations show a complete collapse of the West Antarctic Ice
Sheet by 2300 (Seroussi et al., 2024).

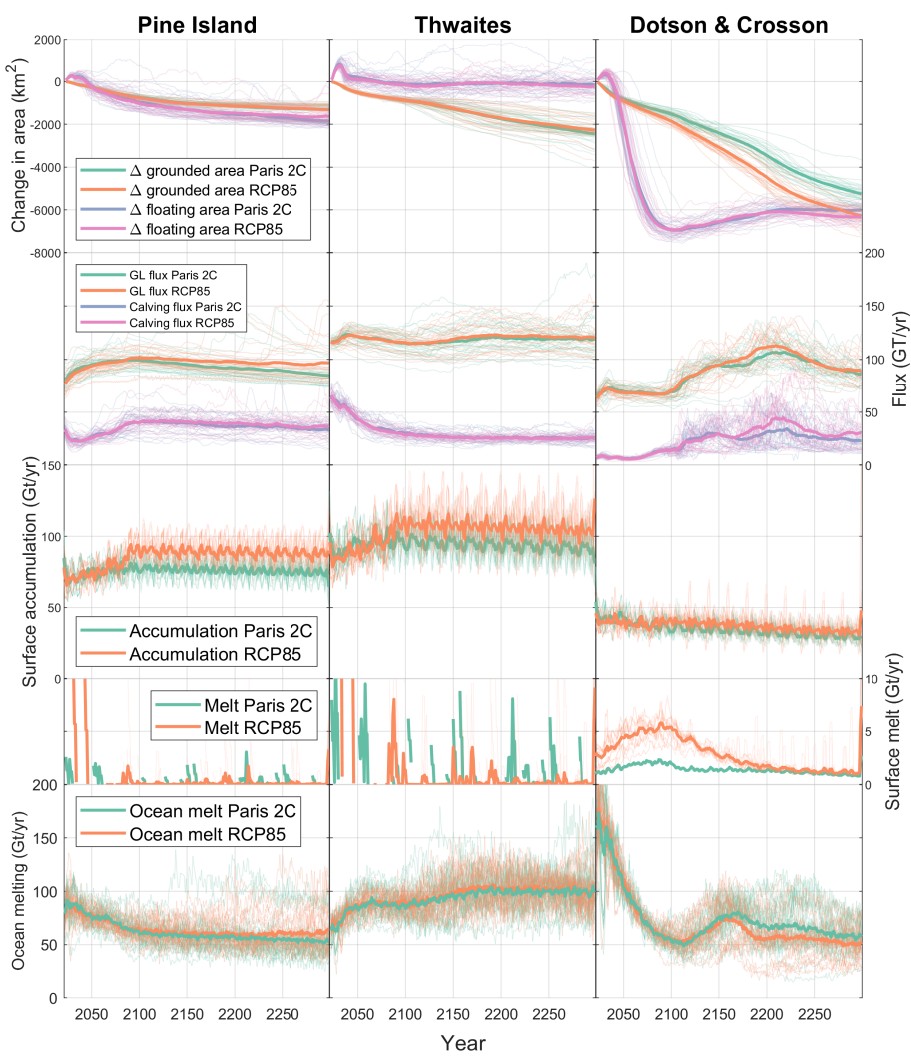

**Figure 10.** Timeseries of model metrics for our simulations extended to the year 2300, showing changes in the area of grounded and floating ice (first row), grounding line and calving flux (second row), surface accumulation (third row), surface melt (fourth row) and ocean induced melt (fifth row). Each metric is calculated by integrating finite element mesh fields over pre-defined catchments consisting of Pine Island (first column), Thwaites (second column) and Dotson & Crosson (third column). Lighter lines show metrics for all simulations individually and heavier lines show the average for each emission scenario (i.e. Paris 2C vs RCP8.5).

## 6 Projections in the context of previous studies

A number of previous ice sheet modelling studies have produced SLR projections for the Amundsen Sea Embayment. Direct comparison with many of these studies is made challenging since they cover a range of time periods, domains and forcings, however limited comparison helps contextualise the results presented here.

The recent study of Wernecke et al. (2020) is arguably the most similar to ours in terms of methodology, using a combination of model emulation and Bayesian calibration of parameters. For 50 year simulations, the spatially calibrated model ensemble resulted in a median of 16.8 mm SLE with 90% confidence intervals of 13.9 and 24.8 mm. Several other studies have produced SLR projections for the region using a model calibrated using Bayesian methods. In Nias et al. (2019), simulations with a duration of 100 years lead to a median SLR contribution of 55.7 mm, increasing to 139.7 mm after 200 years. Even after calibration, there was significant spread in SLR contributions after 200 years, with 5-95th percentiles of 56.2 and 424.3 mm, respectively. In another recent study, Bevan et al. (2023) found a median sea level contribution of 16.2 mm after 50 years, with 5th and 95th percentiles of -0.2 and 39.1 mm, respectively.

A number of other studies have explored the future sea level contribution of the ASE region under various climate scenarios, but without the use of Bayesian methods to calibrate their models. Alevropoulos-Borrill et al. (2020) used a subset of CMIP5 simulations to force model simulations of the ASE and found an SLR contribution of between 20 and 45 mm by 2100 under RCP8.5. A study including the results of 16 ice sheet model found a median SLR contribution for the ASE of ∼20 mm by 2100 under RCP8.5 scenarios with CMIP5 model forcing (Levermann et al., 2020).

The studies described above cover a wide range of timescales, domains and forcings, but in order to make a rough comparison we convert the results listed above to a time-averaged rate and then average these rates across all studies to arrive at a mean rate of sea level contribution of $0.35 \; \mathrm{mm \, yr^{-1}}$, remarkably close to the observed value of 0.37mm/yr for the period 1996-2021 (Davison et al., 2023). Global mean sea levels have risen by $3.7 \pm 0.5 \; \mathrm{mm \, yr^{-1}}$ per year in the period 2006-2018, and further acceleration is considered very likely (Fox-Kemper et al., 2021). So in the context of the recent observed global rate, this averaged rate represents $\sim 10\%$ of the current total. In contrast, in this study we find SLR contribution for RCP8.5 of 0.23 $\mathrm{mm \, yr^{-1}}$ between 2021 and 2100, with a 'very likely' range of 0.14-0.30 $\mathrm{mm \, yr^{-1}}$. Thus, our model projections to 2100 result in a median SLR contribution that is at the low end of the range from previous modelling studies of the region.

Our study differs from the studies outlined above in a number of important aspects. Firstly, we explore uncertainty related to a larger number of model parameters (12), including parameters related to calving, initialisation, ice dynamics, basal and surface mass balance. Secondly, using adaptive mesh refinement on an unstructured mesh allows us to use finer resolution at the grounding line than most other similar studies (400 m in this study). On top of this, by solving velocity and thickness evolution fully implicitly, we expect the simulations presented to provide a more accurate representation of grounding line behaviour. We calibrate our model parameters using observations of changes in both surface elevation and surface ice speed (the model of Bevan et al. (2023) is the only other one to do this for the ASE). Our model simulations also include a dynamically evolving calving front position, rather than a fixed calving front as most previous studies have done. Finally, we use a parameterisation of the plume model to calculate melt rates in response to our evolving cavity geometry and forced by temperature and salinity from an un-coupled ocean model. While a fully-coupled approach would clearly be preferable, this methodology includes more physics than the simple melt rate parameterisations used by other studies that do large ensemble simulations.

Although we cannot directly attribute the reason for our modelled SLR projections being at the low end of previously published results, one plausible explanation is differences in the treatment of surface mass balance. Our study makes use of a PDD model combined with atmospheric forcing from a climate model, including feedbacks in snow accumulation with

660 temperature and surface elevation, that together lead to a strong increase in surface mass balance during the first 80 years of our simulations. In contrast, this feedback is either completely missing or partially absent in all the model studies discussed above, many of which place a heavy emphasis on the response due to ocean induced melting instead (Wernecke et al., 2020; Nias et al., 2019; Bevan et al., 2023; Alevropoulos-Borrill et al., 2020; Levermann et al., 2020).

A weakness of our approach is that we only include results from one ice sheet model and our forcing is derived from only
665 one climate and ocean model, thus a large component of structural uncertainty is not included and would undoubtedly lead to considerably wider confidence intervals than presented here. This is perhaps most notable in terms of ocean forcing, where there is a relatively small spread in thermal forcing arising from the various ensemble members and emissions scenarios which is in contrast to, for example, ASE ocean properties in the ISMIP6 experiments (Fig. A1). In particular, there is little difference between the RCP8.5 and Paris2C MITgcm forcing, which was one of the main findings of that study (Naughten et al., 2023).
Once again, this is in contrast to ocean forcing in ISMIP6 and presumably the similarity in SLR contribution that we find between the two scenarios arises due to our choice of ocean forcing. We have also not included in our uncertainty framework any representation of uncertainties in bedrock geometry which are also likely to contribute to the overall model spread (Sun et al., 2014; Wernecke et al., 2022; Castleman et al., 2022).

Another simplification in our Bayesian approach is to only compare total changes in surface speed and elevation over the
675 entire observational period rather than more detailed temporal variability that has been observed over shorter timescales. While it may initially seem appealing to use as much data as possible, the ability of standalone ice sheet models to capture short term variability in processes such as calving and ocean melting remains poor and so attempting to match higher frequency temporal variability is arguably a mistake at this time. Such an approach could ascribe too high a weight to simulations that are getting the right answer for the wrong reasons and lead to overconfidence in terms of model uncertainty. Along similar lines, since we
do not use a coupled ice-ocean model, our ocean forcing does not change in response to changes in cavity geometry, a process that could play an important role in the evolution of melt rates during our transient simulations (De Rydt and Naughten, 2024). That being said, melt rates calculated by the plume model do change in response to the cavity geometry and so at least part of this feedback is captured by our modelling approach.

Our implementation of the calving laws described in Sect. 2.1.3 is intended to include the possibility of triggering a rapid
MICI type retreat. Increased runoff on ice shelves can induce an increase in the calving rate through the $C_Q$ parameter and parameters related to surface mass balance. If the runoff and calving rate becomes sufficiently large, a rapid 'hydrofracture' driven breakup of the ice shelf could occur, exposing tall ice cliffs which would then rapidly retreat. If this point is reached in a simulation, a MICI tipping point is crossed and rapid retreat occurs without the possibility of stopping until the majority of the drainage basin is ice free. We include this process to compare our results with those of two modelling studies that find very
large SLR contributions of over a metre can occur in 125 years or less (DeConto and Pollard, 2016; DeConto et al., 2021). Although these projections are for all of Antarctica, a large proportion of the modelled mass loss occurs in the ASE region and these SLR projections are generally considerably greater than other modelling studies.

In contrast to the studies of DeConto and Pollard (2016); DeConto et al. (2021), we find no evidence that a MICI type retreat occurs in any of our simulations and thus including the cliff failure calving criterion has no meaningful impact on our SLR

projections. This is in line with another recent study that found no evidence of a MICI retreat in the 21st century (Morlighem et al., 2024). Due at least in part to the absence of a MICI in our simulations, our SLR projections are considerably lower than those of DeConto et al. (2021) for the ASE embayment only, where ∼0.3 m of SLRE mass loss was modelled by 2200 under a +3°C global warming scenario. This is in spite of the fact that a very large proportion of ice shelf area has been lost in most of our model simulations by 2100 (Fig. 6a), implying that loss of these ice shelves has very little affect on future ice loss in the region, as previously shown by Gudmundsson et al. (2023). However, it is important to note that our ocean forcing is held quasi-constant after 2100 and so effectively we can only assess whether tipping points occur for the perturbation up to that date, and cannot rule out the possibility that a continued climate forcing would lead to more significant retreat within a similar timeframe to DeConto et al. (2021). There are also some differences in our modelling approach compared to DeConto et al. (2021); we do not use the flux formula which is known to fail for buttressed ice streams (Reese et al., 2018) and details of the implementation of calving are necessarily different since the model presented here uses finite elements and the level-set method.

## 7    Conclusions

In this study we conduct calibrated ensemble simulations, with forcing derived from a state-of-the-art ocean model and CESM1, yielding SLR projections together with uncertainty estimates for the Amundsen Sea Embayment. We investigate two emissions scenarios: RCP8.5 and Paris 2C, and find the sea level contribution until 2100 to be almost identical for both scenarios (19.3 mm), although with a slightly higher uncertainty for RCP8.5 versus Paris2C ($\pm 2.7$ mm vs $\pm 2.2$ mm). These results are generally slightly lower than previously published SLR projections for the region. We explore how the sea level contribution of the ASE might evolve further into the future by conducting a subset of extended simulations until 2250. Here, the difference between RCP8.5 and Paris2C is slightly larger (median SLR contributions of 6.0 cm vs 7.1 cm, respectively), with Paris2C on average leading to a slightly higher sea level contribution as a result of the increased snow accumulation found in RCP8.5.

The similarity in SLR contribution between the two emissions scenarios that we investigate is unsurprising, given that the major driver of change in this region is the ocean, and both thermocline depth and temperatures from the ocean model that we use are very similar in both scenarios (Naughten et al., 2023). For both scenarios, we find the largest fraction of the total uncertainty comes from parametric uncertainty related to model initialisation, although we do not consider structural uncertainty related to our choice of ocean and atmospheric forcing or the ice sheet model itself. After this, parameters related to ice dynamics and basal melting contribute the most to our calculated uncertainty. Other sources of uncertainty, such as calving and internal climate variability, are relatively small in comparison. The importance of initialisation is particularly noteworthy. Although we carefully calibrate the model to best fit measured changes over the observational record, the initial model state plays an important role in our SLR projections and ongoing work towards new approaches such as time-dependent data assimilation (e.g. Choi et al., 2023) should be a focus for the ice sheet modelling community.

Our model includes the same surface melt and MICI processes as the model of DeConto and Pollard (2016), updated in DeConto et al. (2021). Inclusion of these processes in those studies lead to very substantial and rapid mass loss and these

high-end projections have been the focus of much discussion in the following years. In our simulations, as a result of oceanic melting, calving and increased runoff, the majority of ice shelf area in our simulations is lost by 2100. In spite of this, we find no evidence of a MICI type retreat and in fact the loss of these ice shelves seems to have very little effect on the dynamic mass loss in the region. As a result, our SLR projections by the year 2100 are almost an order of magnitude less than those of DeConto et al. (2021) despite using a more pessimistic emissions scenario.

We have not conducted reversibility experiments that would be required to establish whether any of our simulations have crossed a tipping point with regards to ice loss. While we do not see clear evidence of accelerated ice loss towards the end of our simulations, we do not rule out the possibility that Thwaites or Pine Island Glacier may enter a period of self-enhanced retreat leading to far larger sea level contribution beyond the time scale considered here.

*Code availability.* The ice flow model Úa, used to drive all model simulations in this study, is fully open-source and available at https://zenodo.org/records/10829346 (Gudmundsson, 2024). Code related to uncertainty quantification and training of the surrogate models is available at https://zenodo.org/records/11922614

*Data availability.* Model result files saved every 2 years for our extended simulations, running from 2021 to 2300 for both the RCP8.5 and Paris2C forcing, are available at https://zenodo.org/records/14712131

## Appendix A: Plume model description

### A1 Plume routing algorithm

Plume theory was originally conceived to model a system in one horizontal dimension, a situation for which the source of a plume is clear. In order to adapt plume theory to work for two-dimensional models, Lazeroms et al. (2018) incorporated an algorithm that determines the origin of the plume melting each location on an ice shelf. This is important, because melting at a point is not just a function of local conditions, but also the depth at which the plume originates ($z_{GL}$) and the average slope between the origin and the location being considered ($\theta_b$). Since the work of Lazeroms et al. (2018), further refinements to the plume model parameterisation were described in Burgard et al. (2022), but retaining the original routing algorithm. Here, we describe a modified plume routing algorithm that provides a better fit to melt rate observations in the region, followed by a description of the plume model parameterisation itself.

As a first step, we only calculate ocean-driven melting for nodes considered 'strictly downstream' of a grounding line. This means only for nodes belonging to elements for which no other node is upstream of a grounding line, to avoid ocean-driven melting leaking upstream of the grounding line Note that some recent studies suggest ocean water may indeed intrude upstream of the grounding line (e.g. Bradley and Hewitt, 2024; Rignot et al., 2024) and establishing where and when this process may be important is crucial for future ice sheet modelling studies due to their sensitivity to melting in this region. For each of these

ice shelf nodes, the routing algorithm must determine an origin location for the plume that then drives melting at that point. A list of candidate nodes is generated from all nodes along the main ice sheet grounding line, meaning that plumes cannot originate from pinning points or inland subglacial lakes. For each list of nodes for which a melt rate must be calculated and

760 candidate plume origin nodes, we create an array of $x$, $y$ and $z$ coordinates (where the $z$ coordinate is given by the ice shelf draft $b$ for each node), and we refer to these arrays as $N_m$ and $N_{GL}$ respectively. The $z$ coordinate of $N_m$ is then offset by a large negative number and $z$ coordinate of $N_{GL}$ is multiplied by two. Finally, we use a standard k-nearest neighbour algorithm to find the $k$ nearest points in $N_{GL}$ for each point in $N_m$. The effect of the modifications to the $z$ coordinate is to make deeper candidate nodes in $N_{GL}$ appear closer to nodes in $N_m$ and therefore preferred as plume origin nodes. We choose $k = 10$ for

our simulations, leading to 10 candidate plume origin nodes for each node in $N_m$.

We calculate the depth of the origin plume ($z_{GL}$) for each melting node as the average ice draft of the $k$ plume origin nodes. The average slope ($\theta_b$) at melting node $i$ is calculated as

$$\theta_{b,i} = \sum_{n=1}^{n=k} \frac{b_i - b_{i,n}}{L_{i,n}}, \tag{A1}$$

where $b_i$ is the ice draft at node $i$, $b_{i,n}$ is the ice draft at the plume origin node $n$ and $L_{i,n}$ is the horizontal Euclidian distance

between node $i$ and plume origin node $n$.

In addition to the non-local quantities calculated above via the plume routing algorithm, melt rate at each node is a function of local slope ($\theta_l$), ice draft ($b$), and a depth dependent local temperature and salinity. Vertical profiles of temperature and salinity are generated as a function of ocean model thermocline depth ($z_T$) and temperature and salinity averaged above ($T_s, S_s$) and below ($T_d, S_d$) the thermocline as follows:

$$T(z) = \begin{cases} T_s + b \left( \frac{T_d - T_s}{z_T} \right) & \text{if } z \geq z_T \\ T_d & \text{otherwise}, \end{cases} \tag{A2}$$

resulting in a linear variation in temperature (salinity) from the base of the thermocline ($b = z_T$) where $T = T_d$ to the surface ($b = 0$) where $T = T_s$. These quantities are extracted from MITgcm for each basin (Pine Island, Thwaites and Dotson/Crosson), and the resulting values are shown in Fig. A1 including the range of internal climate variability ($E_{ID}$) and emissions scenario. Also shown for comparison are ocean temperature extracted with the same method from the ISMIP6-2300 experiment forcing

(Seroussi et al., 2024; Nowicki and Team, 2024) for contrasting scenarios and different climate models (although our method to calculate melt rates from the ocean forcing differs significantly from those experiments).

Finally, the plume parameterisation requires a mean temperature of water entering the cavity ($\bar{T}$), which is calculated as the weighted average of temperatures in the cavity for each node i.e.

$$\bar{T} = \begin{cases} (T_z + T_d)/2 & \text{if } z_{GL} \geq z_T \\ T_d & \text{if } b \leq z_T \\ \frac{(b - z_T)(T_z + T_d)/2 + (z_T - z_{GL})T_d}{b - z_{GL}} & \text{otherwise}, \end{cases} \tag{A3}$$

where $T_z$ is the local temperature at the ice base $T(z = b)$.

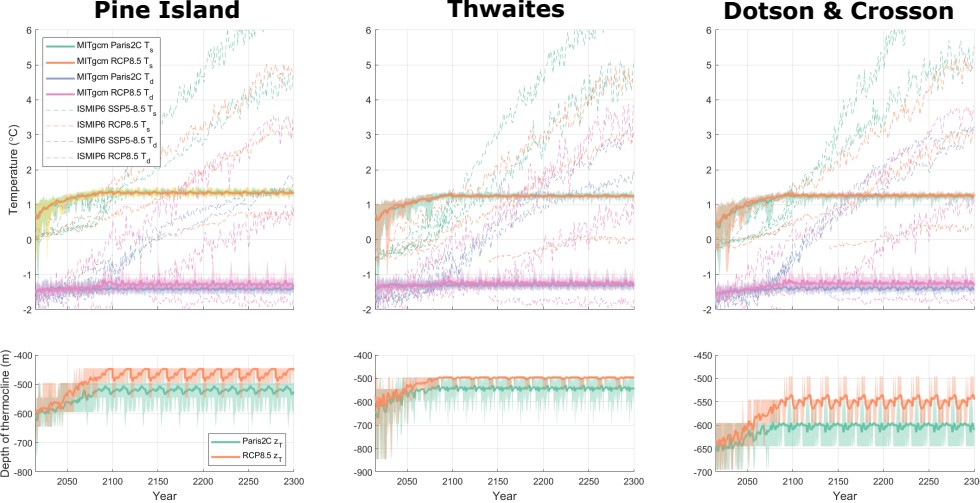

**Figure A1.** Ocean forcing in the three catchments of our model, extracted from MITgcm simulations for the RCP8.5 and Paris2C scenarios (Naughten et al., 2023). The shaded area shows the range for each value emerging from different realizations of internal climate variability $E_{ID}$, while the solid line shows the ensemble mean. Also shown are temperatures extracted from ISMIP6-2300 ocean forcing for different climate models and the SSP5-8.5 and RCP8.5 scenarios (Nowicki and Team, 2024; Seroussi et al., 2024). The temperatures plotted are extracted from all model outputs in the same way, as described in Sect. 2.2. the temperatures ($T_s$, $T_d$) above and below the thermocline ($z_T$) yield a vertical temperature profile as defined in Eq. A3.

## A2 Ocean induced melt rate calculation

To calculate the melt rate, we implement the plume parameterisation of Lazeroms et al. (2018) in a modified form that partially follows Burgard et al. (2022). We briefly summarise the equations as there are some subtle differences in the definition and use of average properties required by our revised plume routing algorithm (A1).

Using the unmodified Lazeroms et al. (2018) parameterisation the melt rate can be expressed as a function of scaled distance from the grounding line, $x'$:

$$\dot{m} = \left( \frac{C_d^{1/2} \Gamma_{TS}}{L/c} \right) \left[ U' f_U(x') \right] \left[ \Delta T' f_{\Delta T}(x') \right] \tag{A4}$$

where the velocity and thermal forcing scales are:

$$U' = \left[ \frac{\beta_T g E_0 \sin \alpha}{\lambda_3 (C_d + E_0 \sin \alpha)} \right]^{\frac{1}{2}} \left\{ \frac{C_d^{1/2} \Gamma_{TS} \left[ \frac{\beta_S}{\beta_T} \left( \frac{S_a c}{L} \right) - 1 \right]}{C_d^{1/2} \Gamma_{TS} \left[ 1 - \lambda_1 \left( \frac{S_a c}{L} \right) \right] + E_0 \sin \alpha} \right\}^{\frac{1}{2}} (T_a - T_{f,gl}) \tag{A5}$$

$$\Delta T' = \left[ \frac{E_0 \sin\alpha}{C_d^{1/2}\Gamma_{TS}\left[1 - \lambda_1\left(\frac{S_a c}{L}\right)\right] + E_0 \sin\alpha} \right] (T_a - T_{f,gl}) \tag{A6}$$

the dimensionless velocity and thermal forcing functions are given by:

$$f_U(x') = \frac{1}{\sqrt{2}}(1-x')^{\frac{1}{3}}\left[1 - (1-x')^{\frac{4}{3}}\right]^{\frac{1}{2}} \tag{A7}$$

$$f_T(x') = \frac{1}{2}\left[3(1-x') - \frac{1}{(1-x')^{\frac{1}{3}}}\right] \tag{A8}$$

and the scaled distance is defined as:

$$x' = \frac{\lambda_3(z - z_{gl})}{(T_a - T_{f,gl})}\left\{1 + 0.6\left[\frac{E_0 \sin\alpha}{C_d^{1/2}\Gamma_{TS}\left[1 - \lambda_1\left(\frac{S_a c}{L}\right)\right] + E_0 \sin\alpha}\right]^{\frac{3}{4}}\right\}^{-1} \tag{A9}$$

In the original derivation of the above equations the ambient ocean properties ($T_a$, $S_a$) are considered uniform, while the 1-d ice shelf base shallows monotonically with a constant basal slope ($\sin\alpha$) from its maximum depth at the grounding line depth ($z_{GL}$). Lazeroms et al. (2018) demonstrated that the parameterisation could be extended to non-uniform slopes simply by using the local value of slope in the above scales, while Burgard et al. (2022) introduced further modifications to account for depth variation in the ambient ocean properties. Here, we follow Lazeroms et al. (2018) in using local slope, except in the definition of the scaled distance ($x'$), where we make use of the mean slope defined in A1. In the length scale definition, we further make use of the mean grounding line depth and mean ambient ocean temperature, also defined in A1. We then follow Burgard et al. (2022), in making use of the mean temperature in the expression for $\Delta T'$ and the local temperature to calculate $U'$. Note that, our plume routing algorithm focuses the origins along deeper parts of the grounding line, unlike the uniform sampling of the cavity inherent in the original algorithm of Lazeroms et al. (2018) that was subsequently used by Burgard et al. (2022). We therefore do not make use of a cavity wide average of either temperature or slope. In Fig A2 we show basal melt rates varying in response to forcing and cavity geometry for an example simulation, as calculated by the plume parameterisation described above at various times between 2021 and 2300.

## Appendix B: Parameter Priors

As a first step to evaluating a posterior probability we require priors for our selected parameters of the Ice Sheet, plume and PDD models (referred to as our three forward sub-models hereafter). One choice would be to use uniform (uninformed) priors, or to base our priors on some distribution of the parameter choice in previous studies (e.g. Hill et al., 2021). However this does not take advantage of the extensive observations that exist in the ASE, particularly since the late 1990s when the satellite

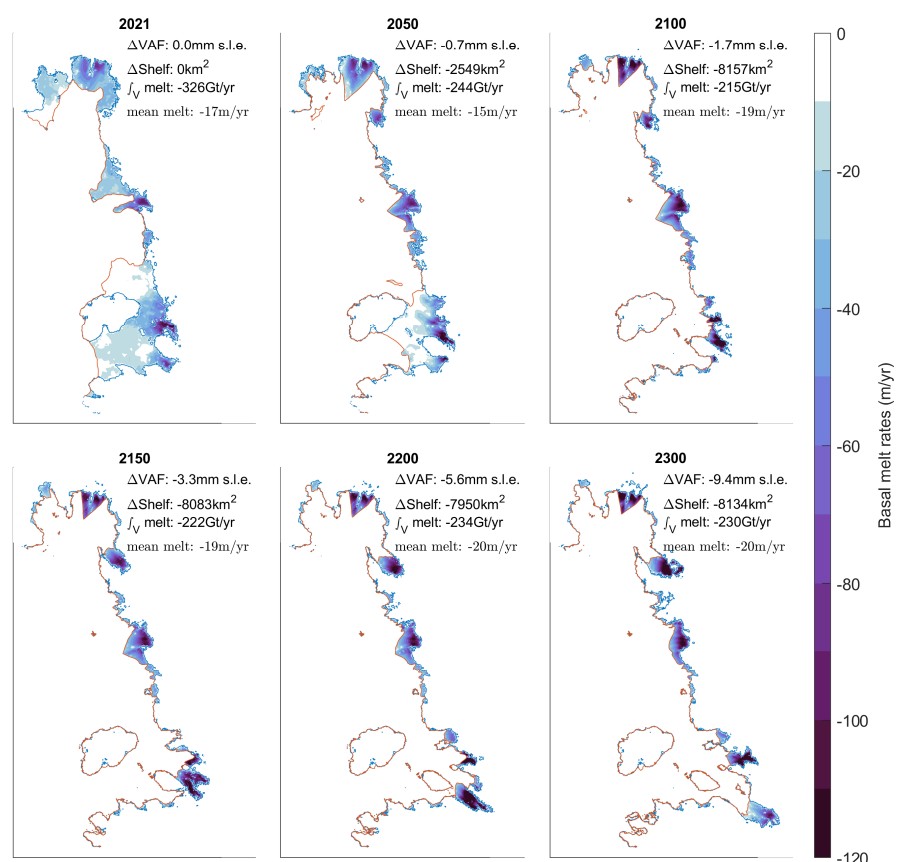

**Figure A2.** Basal melt rates at 6 time slices for a single extended Paris 2C simulation, whose final change in Volume above flotation by 2300 was closest to the median of all Paris 2C simulations. Solid red line is the calving front position and solid blue line is the grounding line. $\Delta$VAF and $\Delta$Shelf are change in volume above flotation and change in area of the floating ice shelf with respect to the modelled 2021 values. $\int_V melt$ is the integrated basal melt rates across the whole domain and mean melt is total melt divided by ice shelf area.

record becomes more complete. In the Bayesian framework, these observations can be used to update our probabilistic beliefs. More specifically, we can calculate the posterior probability for our model parameters, given the observations, and then use this posterior as our prior for the model parameters in the future simulations.

## B1   Model inversion priors

Our forward model is initialised using inverse methods so that our simulations agree with observed velocity and surface elevation changes at their starting date. As described in Sec. 2.1.4, this results in 5 uncertain model parameters related to regularisation of the inverted $A$ and $C$ fields ($\gamma_{aC}$, $\gamma_{aA}$, $\gamma_{sC}$ and $\gamma_{sA}$) and the relative weighting of the velocity observations versus the surface elevation change observations ($\dot{h}_e$). A standard approach when selecting the magnitude of these regularization pa-

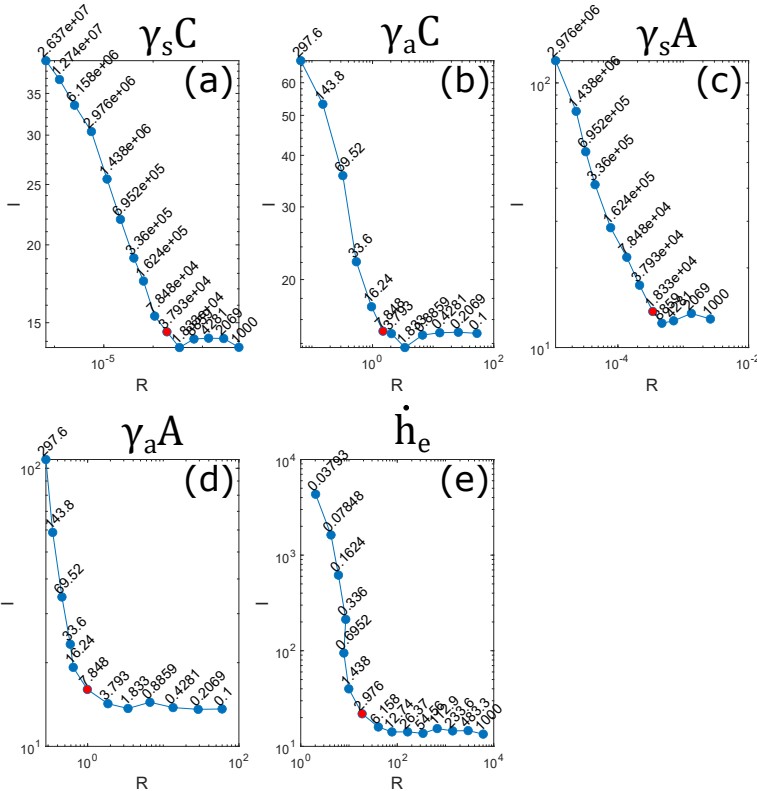

**Figure B1.** L-curve analysis for model initialisation hyperparameters, showing change in the regularisation term ($R$, Eq. 14) vs. change in misfit term ($I$, Eq. 15) for different values of the regularization parameters (where the value of the parameter is given as a label on each point in the plot.

rameters is to conduct an L-curve analysis. The motivation behind this is that there exists a tradeoff between the size of the regularization term ($R$, Eq. 14) and how well the resulting model agrees with the data (i.e. the misfit term, $I$ in Eq. 15). By running the inversion for different amounts of regularization and creating a log-log plot of these two terms, an optimal balance

can be determined as the point of greatest curvature in the plot. We conducted this analysis for the five parameters and the L-curves for each one is plotted in Fig. B1.

To define the prior probability distributions for each inverse model parameter we selected Gaussian distributions whose mean is the optimal point determined by the L-curve analysis (red points in Fig. B1). It is not possible to attribute a variance based on the L-curve plots and instead we assign a value such that 99.75% of the distribution falls within one order of magnitude

either side of this mean. This conservative estimate is based on the fact that the L-curve analysis is an ad hoc strategy to tune the regularization parameters and so the most appropriate values may lie some distance away from the points selected by this method.

## B2 Plume model priors

We assign prior probabilities to the two plume model parameters based on how well the plume parameterisation matches with
melt rates as simulated by the MITgcm ocean model simulations of Naughten et al. (2023). This additional Bayesian inference
step uses the same framework as described in Sec. 2.4.3 and the posterior probabilities calculated as described in this section
can then be used as prior probabilities for the main model calibration. This ensures that the plume model parameters used in
our forward simulations are broadly consistent with the MITgcm model that drives our ocean forcing while also leading to
changes in ice velocity and surface elevation in the ASE that are consistent with observations. To do this, we must first define
priors for the plume model parameter calibration that follows. We choose Normal distributions whose mean is centered around
the values proposed by (Lazeroms et al., 2018) and with a standard deviation such that 99.7% of the distribution lies within
one order of magnitude of the mean.

     Since the MITgcm simulations do not include feedbacks on the ice shelf geometry, we run the plume model in a standalone
configuration, with the same ice shelf cavity geometry as prescribed in Naughten et al. (2023) and driven by changes in
ocean temperature, salinity and stratification provided by the same model. This is done for all model ensemble members and
for both the RCP8.5 and Paris 2C scenarios. As outlined in Sec. 2.4.1 and with the same motivation, we greatly reduce the
dimensionality of the melt rate fields that we compare by constructing a SVD of the plume model melt rates. We find that a
truncation of $k = 10$ enables us to account for $> 95\%$ of the variance in a large ensemble of standalone plume model simulations
spanning the parameter space. In this case, however, the standalone plume model is sufficiently computationally efficient that
we do not require a surrogate model. Other than this, the calculation of posterior probabilities follows the same procedure
as in Sec. 2.4.3, whereby the likelihood is the difference in modelled melt rates in principal component space (Eq. 24). The
maximum a posteriori estimates resulting from this analysis are $\Gamma_{TS} = 1.75 \times 10^{-4}$ and $E_0 = 2.2 \times 10^{-2}$, compared with
previously published values of $\Gamma_{TS} = 6.0 \times 10^{-4}$ and $E_0 = 3.6 \times 10^{-2}$, given by Lazeroms et al. (2018).

## B3 Ice-flow model priors

We include two ice-flow parameters in our uncertainty analysis; related to the ice rheology ($n$, Eq. 1) and the basal sliding
($m$, Eq. 2). Historically, these have typically both been set equal to three but increasingly this paradigm is being challenged.
In the case of Pine Island Glacier for, example, two recent studies have found that using a more strongly nonlinear basal
sliding parameter is more consistent with observations (Gillet-Chaulet et al., 2016; De Rydt et al., 2021). Similarly, increasing
availability and accuracy of remote sensing data along with new methodologies have called into the modelling community's
adoption of $n = 3$, with some lines of evidence suggesting a greater sensitivity of ice viscosity to stress is necessary (Millstein
et al., 2022; Wang et al., 2022). Despite an increasing amount of research in both these parameters, no clear consensus has
emerged on the best choice in either case. For this reason, we choose uniform priors for both, bracketed by upper and lower
bounds of $n = 2$ to $n = 5$ and $m = 2$ to $m = 8$. Note that at the upper end of our prior for $m$, basal sliding becomes effectively
plastic.

## Appendix C: Un-calibrated model parameters

Parameters of the PDD model, whose role is to include changes in surface mass balance under warming scenarios, cannot be readily calibrated on present-day ASE observations where these processes are currently very limited. Similarly, the parameter $C_Q$ controlling the sensitivity of calving rate to surface meltwater is included to introduce the possibility that increased atmospheric warming could lead to accelerated ice shelf collapse but this process has not been observed in the region. Without relevant observations or in absence of sensitivity to these parameters under present-day conditions, assigning posterior probabilities is not possible as it is with the other parameters in Table. 1.

The $\sigma_T$ parameter of the PDD model largely determines the amount of surface meltwater produced as a result of changes in temperature and thus we can use future simulations of the Regional Atmospheric Model (MAR), that include this process at a point in the future where it becomes relevant, to provide a posterior probability. For this purpose, the PDD model can be decoupled from the ice sheet model, driven by atmospheric forcing and solving for meltwater production with no feedbacks on ice sheet geometry. A recently published modelling study ran the MAR until the year 2100 for the Amundsen Sector, driven by CMIP5 anomalies for the RCP8.5 emissions scenario and found an order of magnitude increase in production of surface meltwater on ice shelves in the region (Donat-Magnin et al., 2021; Marion et al., 2019). By using the same atmospheric forcing as the MAR simulations and comparing meltwater production calculated by the two models in the final twenty years (2080-2100), we can constrain $\sigma_T$ by finding the value that leads to the best agreement with the state-of-the-art MAR model. Defining the decoupled PDD model as our forward model and the MAR modelled meltwater production as our observations, we can calculate posterior probability using the same framework described in Sec. 2.4.3. We repeated this analysis using $\sigma_M = const$ as a replacement for Eq. 6 but found that this provided a less good fit to the MAR model data.

A different approach is required for the remaining two parameters, $p$ and $C_Q$, where analogous model simulations in the region that can inform our parameter choices are not readily available. As an alternative, we use published values in the literature to quantify uncertainty. The parameter $p$, representing the sensitivity of precipitation to changes in air temperature, is widely used within the ice-sheet modelling community and a recently published study by (Nicola et al., 2023) provides an overview of published from a variety of models. We use these values to define a Gaussian prior with mean and standard deviation calculated directly from the spread of previously published values. The parameter $C_Q$ is unique to the recent study by DeConto et al. (2021), who vary this parameter between 0 and $195 m^{-1} yr^2$ and we adopt the same conservative approach, sampling this parameter from a uniform distribution with the same limits.

## Appendix D: RNN surrogate

As outlined in Sec. 2.4.2, we require a computationally cheap surrogate model to map from any given $\theta$ to a prediction in principal component space ($\tilde{V}^T$) in our Bayesian inference. We choose to use a residual neural network as our surrogate model, due to its simplicity and ability to learn complex nonlinear relationships within the training set. Our goal is that, once training is complete, our surrogate model can predict changes in surface ice speed and elevation over the duration of our chosen observational period for a previously unseen combination of model parameters.

As a first step, we define a flexible architecture consisting of an input layer, a series of $N$ residual blocks, and an output layer. Each residual block consists of a series of $D$ sequences of a fully connected layer with $H$ neurons, a batch normalization layer and dropout layer. The output of a residual block is given by the sum of the output of the last dropout layer in the sequence and the its input. In all fully connected layers apart from in the output layer we use swish activation functions. The loss function, defined as the Huber loss between modelled and predicted speed and surface elevation change and reprojected from the reduced SVD representation to the model space, is minimized with the Adagrad optimizer.

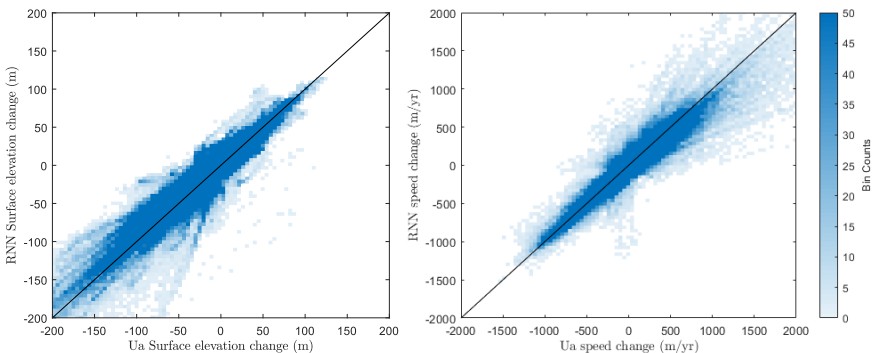

**Figure D1.** Change in surface elevation and ice surface speed between 1998 and 2021 as predicted by the Úa-*obs* simulations and the RNN surrogate model (2.4.2).

We train this network using a large ensemble of 5670 model simulations with sampled model parameters $\Theta$ and modelled surface ice speed and surface elevation change, reprojected using Eq.18. We split our $m$ model simulations into training, validation and test sets with proportions of 70%, 15% and 15%, respectively. The model is built using Tensorflow and network optimization is automated using the Bayesian optimization option in the Keras tuner module to find a learning rate $l_R$, depth $D$, number of residual blocks $N$, number of neurons in each layer $H$ and dropout fraction $d$ that yield the best validation loss. We run this optimization with various choices for $K$, so that the choice in truncation of the SVD is informed by what results in the best surrogate model. Our final network used $D = 1$, $N = 4$, $H = 115$, $l_R = 0.25$, $d = 0.1$ and $K = 0.9$ (resulting in a trunctation of $k = 8$ for surface elevation change and $k = 11$ for change in speed. Fig. D1 shows the results of the network evaluated on the test set, at every node in the model domain, showing good agreement between the surrogate model and the forward model with an RMSE of 4.77 m and 35.33 m yr$^{-1}$ for surface elevation and speed change, respectively.

## Appendix E: LSTM surrogate

We train a surrogate model to make predictions of SLR contribution for a combination of model parameters and forcing ensemble member. Although our primary motivation is to obtain projections for the year 2100, we can make better use of our training simulations by leveraging the inherent time dependency and so the network is trained to make predictions for every year between 2021 and 2100. For this purpose, a Long short-term memory (LSTM) model is appropriate, as it can learn longer

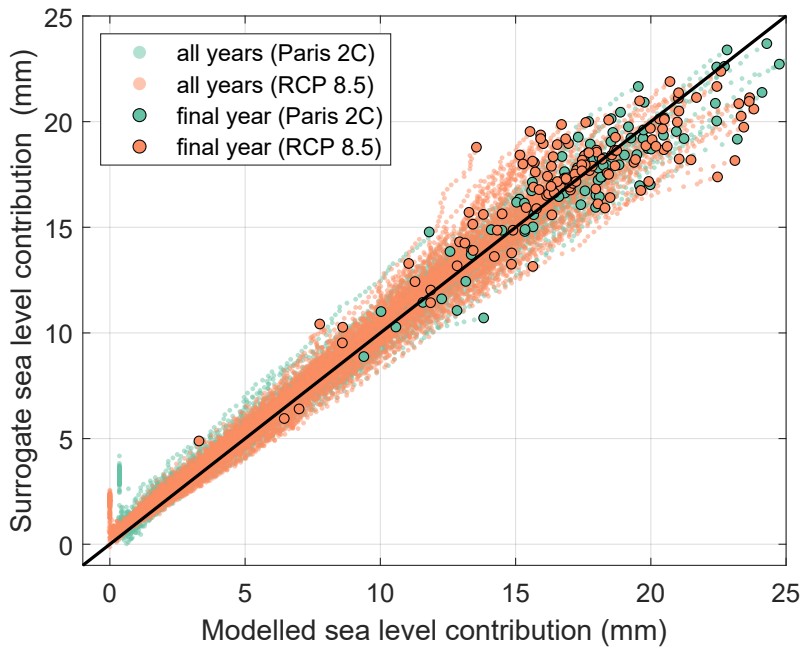

**Figure E1.** Cumulative sea level contribution from the ASE for each year, between 2021-2100, as predicted for the test subset of Úa-*fwd* simulations and the LSTM surrogate model. The cumulative sea level contribution for the final year of each simulation, which make up the bulk of our uncertainty quantification and analysis, are highlighted with a black edge.

time dependencies than a standard recurrent neural network. An LSTM unit consists of three gates that regulate the flow of
925 information in and out of its current state (analogous to the network's memory). Predictions are made sequentially, based on
the input to the LSTM and its current state. The first layer of the network is a sequential input layer that takes normalized input
parameters from a training subset of the Úa-*fwd* simulations. These are passed to a series of blocks consisting of an LSTM unit
and a dropout layer. Finally, a fully connected layer takes the output of the final block and returns a scalar value representing
sea level contribution for each time interval that the model is run for. The network loss was calculated using the Huber loss
function on the predicted timeseries compared to the result calculated from the Úa-*fwd* simulations. As with the surrogate
described in Sec. 2.4.2 we optimized the networks hyperparameters to minimize the validation loss, resulting in a network with
2 blocks of 34 hidden units, a dropout of 0.5, an initial learning rate of 0.007 that was reduced every 10 epochs and a mini
batch size of 32. The network was trained separately for the two scenarios using 1183 and 1107 simulations for the Paris2C
and RCP8.5 simulations, respectively, of which 20% were reserved for testing and validation. Fig E1 shows the performance
of the two trained surrogate models on the test set of each scenario, compared to the outputs from the Úa-*fwd* simulations.

*Author contributions.* SHRR designed and carried out all model experiments and sensitivity analysis and produced the figures and paper. GHG developed the ice sheet model and secured funding. AJ helped to develop the basal melt algorithm and contributed relevant sections to the paper. KN provided atmospheric and ocean forcing.

*Competing interests.* The authors declare that they have no conflict of interest.

*Acknowledgements.* This work is from the PROPHET project, a component of the International Thwaites Glacier Collaboration (ITGC). Support from National Science Foundation (NSF: Grant 1739031) and Natural Environment Research Council (NERC: Grant NE/S006745/1). Logistics provided by NSF-U.S. Antarctic Program and NERC-British Antarctic Survey. ITGC Contribution No. ITGC-129.

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
