# Peer review of "Calibrated sea level contribution from the Amundsen Sea sector, West Antarctica, under RCP8.5 and Paris 2C scenarios"

_EGUsphere, 2024_

## Referee Comment (RC3)

**Title**: Calibrated sea level contribution from the Amundsen Sea sector, West Antarctica, under RCP8.5 and Paris 2C scenarios
**Authors**: Roiser et al. (2024)
**Journal**: The-Cryosphere

**Overview**:
In this research article, Roiser et al. present numerical ice sheet modeling results of the Amundsen Sea Embayment (ASE), a dynamic region of West Antarctica that is rapidly contributing to global sea level, to 2100 and 2250 under RCP8.5 and Paris-2C emission scenarios. Central to the paper is the development of a new framework for quantifying uncertainties associated with these modeling results by training a surrogate model. Overall, the authors find that the sea level contribution of the ASE through 2100 is nearly identical in both RCP8.5 and Paris-2C, tending to the lower range of other published sea level projections of this region. Grounding line migration is minimal through 2100; however, iceberg calving drives near-total loss of all floating ice shelves. Beyond 2100, the simulations diverge as snowfall ramps up in RCP8.5. In terms of the uncertainty quantification, the authors found that parameters related to model initialization (e.g., hE and the inversion coefficients) comprised a majority of the uncertainty, followed by ice flow parameters, basal mass balance parameters, and surface mass balance parameters.

Overall, I find this manuscript to be of very high quality and I expect the results will be of wide interest to the glaciological and cryosphere community. I also believe that the methods of uncertainty quantification are exceptional and provide critical insight into ice sheet modeling parameters. I do share some similar concerns with the other reviewers in that I would like to see further interpretation of the forward simulation results. In particular, I find it very interesting that there is minimal grounding line retreat but near-complete loss of floating ice shelves through 2100 – I would like to see the authors dig into the reasons for this as well as into the reason why their estimates are on the lower end of published sea level estimates of this region. I also have a note in the general comments section that I think it would be appropriate for at least a subset of the ice sheet modeling results to be made publicly available given the broad interest in this region. Lastly, I also have a couple more minor line comments that should be easily addressed. Once these issues are addressed, I would be very happy to support prompt publication of this work in The-Cryosphere.

**General Comments**:
- **Additional discussion related to modeling results**: Like the other reviewers highlighted, I think that the results of this paper can be bolstered by diving deeper into analysis of the Ua_fwd projections at both 2100 and 2250. In particular, the authors present a unique implementation of the buoyant plume parameterization and a new way of prescribing oceanic temperature and salinity inputs to this parameterization (using the modeled depth of the thermocline as a depth-cutoff in applying T/S). Given that this region of Antarctica is primarily forced by the ocean, I would like to see how these modeled melt rates compare to contemporary estimates and how they change in time with evolving cavity geometry. I also think that comparisons of how much ice mass is added by atmospheric processes versus how much ice mass is advected across the grounding line in the different simulations would be helpful to diagnose why mass loss is generally low in these simulations. Lastly, as I mentioned below in the line comments, I would

like to see the introduction of the manuscript expanded to provide an overview of ASE modeling efforts.

- **Some consistent formatting fixes**: Throughout the manuscript, I noticed that in-text citations were not formatted correctly (e.g., in-text citations should be Lazeroms et al. (2018) and not Lazeroms et al. 2018). Also, there are many times in the manuscript where there needs to be a space between the value and the unit (e.g., 1mm should be 1 mm).
- **Data Availability**: While the authors state that no new datasets were used in this article, they did generate ice sheet numerical modeling outputs that would be of great interest to the cryosphere community given how active numerical modeling efforts are in this area. While I understand that depositing all of the data might be challenging since there are so many model runs, I would implore the authors to deposit a representative subset of the results (or at least the data needed to replicate the figures) in a publicly curated data repository.

**Line Comments**:
- Abstract: Much of the paper focuses on the development and implementation of the surrogate model and Bayesian calibration, so I was surprised to see that there was no mention of this in the abstract. If there is room, I would add a sentence about this.
- L17: Capitalize "Pine Island"
- L20: Change "rate" to "rates"
- L27: What do you mean by " . . . the complex response of the ocean and ice shelf cavities to atmospheric forcing"? Is this referring to changes in wind stress that impact ocean circulation? For the ice shelves, does this mean enhanced surface melt that might lead to hydro-fracture? Are there studies that you can highlight that show the connection between atmospheric circulation and sub-ice shelf ocean cavity circulation for ASE glaciers? It would be good to specify and connect this to the next sentence because it feels a little ambiguous and out of place currently.
- L37: Please add a citation for the plume model.
- L15:42: I was hoping to get a little more background on ASE glaciers and recent ice-ocean model studies of this region. I think this will be particularly important if you choose to include additional discussion regarding the results of your ASE sea level rise projections. For example, this would be a great place to discuss what sea level projections of this region exist, what are their shortcomings, what are the partitions of sea level contribution between Thwaites and Pine Island, what ocean and atmospheric forcing datasets and parameterization have been used, have tipping points been found in previous studies (note that these are examples and you can select what you think would be appropriate)? A bit more background information to help put your research in the context of what has been done already would be very useful.
- L44: Add a comma after "uncertainties"
- L45: Change comma to a period after "together"
- L62: What was the minimum and maximum mesh resolution?
- L65: Please add a citation for Glen's flow law.
- L72: Please add a citation for the friction law. Also, what does $m$ control in this friction law?
- L100: Where did the value of 1.66 come from?
- L127: ". . . in a coordinate system X oriented along the . . ."
- L138: What was the reasoning for picking $C_d^{1/2}\Gamma_{TS}$ as the uncertain model parameter rather than $C_{d}$, the ice-ocean drag coefficient?

- L142: Remove "of"
- L148: Was there a specific reason you left out melting from geothermal heat flux? It seems like this would be a relatively simple addition to the model since you can assume it stays constant through 2100. I don't think this would be a reason to re-run the simulations, but I am just curious.
- L151: Check the formatting of your references here – I think they should be "Pollard et al. (2015) and DeConto et al. (2021)
- L158: General comment about units, make sure to add a space between the value and the unit (e.g., 1.5 m/yr rather than 1.5m/yr). I saw this come up in a couple of lines now. The only unit that should be directly against the value is the degree-symbol.
- L174-175: Citation should not have parentheses
- L200: What version of BedMachine do you use?
- L206: What specifically are these assumptions? Do you assume floating ice is at hydrostatic equilibrium? If so, please state this and other assumptions explicitly.
- L235-237: This description of how ocean T/S is applied to the plume model is a bit vague. So within each of the three major ice shelf cavities (Pine Island, Thwaites, and Crosson/Dotson), you extract the thermocline depth and T/S above and below that depth. So you get 15 values (5 for each of the three ice shelves), but how is that T and S applied within the plume model? At all ice shelf depths above the thermocline, you apply the average T above the thermocline, and vice versa for below the thermocline? Is this done at every time-step? How does the location where you obtain these T/S/depth variables change as the ice shelf cavity geometry changes in time (i.e., do you always sample from the same locations, or does the location follow migration of the grounding line and ice front)?
- L415: It might be nice in this section to also mention the rate of sea level rise of the ASE, not just the final 2100 contribution. In figure-5, it looks like the rate of sea level contribution from RCP8.5 starts to decrease after 2070, do you know why that is?
- L435-455: I agree with the other reviewers that further discussion of these results would be really valuable! In particular, it is quite surprising that you see little grounding line retreat across both Pine Island and Thwaites Glaciers – some analysis on the ice shelf melt rates that were applied throughout these simulations would be very helpful. Perhaps you could show what the melt rates look like at a couple of different time-steps? You could also show a time-series of total integrated ice shelf basal melt across the main shelves in the domain.
- L451 In figure-6, do you know why Pine Island Ice Shelf ends up shaped like a triangle at both 2100 and 2250? I am very surprised to see that this same ice front shape holds throughout the whole simulation.
- L478: Remove "in the" once; you have it written twice consecutively in this line.
- L481: This comparison to past results is a great starting point! I do think that you should go a step farther and try to deduce why your modeled sea level contributions are on the low end of other published results. Like I said before, I think it is worth looking at what your ice shelf basal melt rates look like in the Ua_fwd simulations and how they change in time (a figure of this would be valuable, maybe in an appendix or supplement). Another possible cause of limited mass loss could be the treatment of surface mass balance, which you could look into by quantifying how much snow fell over the simulation period and how much this offsets ice loss at the grounding line. Do the other projections you cited include atmospheric forcing as a positive degree day

parameterization, or do they directly apply CMIP anomalies in SMB within their model without correction?

- L495: Is this statement about your melt rates true? I thought that the plume parameterization updates ice shelf melt based on changing ice shelf basal slopes and the depth of the grounding line? Maybe it would be more accurate to say that you cannot resolve 3-dimensional sub-ice shelf ocean circulation.
- L505: Might be worth comparing to the new Science Advances paper by Prof. Morlighem (https://www.science.org/doi/10.1126/sciadv.ado7794). They do not find any evidence for MICI over the next 50 years, so this supports your findings well.
- L525-530: A similar analysis of results for your 2100 simulations might be helpful in determining why mass loss is on the lower end of published projections.
- L533-535: Can you determine or at least speculate on why there is a decrease in RCP8.5 mass loss prior to 2100?
- L541: The sea level contribution is similar up until 2100, but not through 2250. Please specify this.
- L560: Check the format of the citation, should have parentheses.

**Figure Comments**:
- Figure 1: This is a very helpful figure and I like that you included the associated section and appendix numbers in each of the boxes. In the figure caption, can you please include the definition of acronyms (e.g., RNN, LSTM, and Del GMSL).
- Figure 2/3: Should there be a color bar associated with this figure? Also, is the top row showing surface ice speed, or the change in surface ice speed (as would be consistent with a divergent colorbar)? This last comment applies for figure-3 as well – should this say change in surface ice speed? Lastly, for figure-3, why are there no changes shown over floating ice?

---

## Author Comment (AC1)

We thank the reviewer for their time and effort in providing feedback on our manuscript. Our responses to their comments are given below, in red. In the case where no direct response is given, we will implement the reviewer's suggestions directly into the revised manuscript without further change.

**General comments**

This paper presents a thorough and rigorous method for producing calibrated projections of sea level rise from the Amundsen Sea sector of West Antarctica, using a multistep process to consider and constrain uncertainty from a wide range of sources, by combining both a numerical ice flow model and a statistical surrogate model. The resulting sea level contributions by 2100 under RCP8.5 and Paris2C are at the lower end of the range of estimates from previous studies and similar between the two scenarios. They also extended some simulations to 2250 and found the two scenarios diverged after 2100 due to differences in snow accumulation.

In general, the study is very well presented (save a few consistency errors – see technical corrections below) and structured clearly, with a lot of methodological detail. However, I think certain later sections could benefit from further discussion and interpretation, e.g. Section 4, in particular, is very brief and offers no real interpretation, which is a bit of a shame – see specific comments for more detail.

All reviewers agreed that further discussion and interpretation of the results would be beneficial to the paper and we will expand on numerous sections based on this feedback.

**Specific comments**

There are several places where statements are made without supporting citations:

- L83-84: give examples of studies that use a mixed sliding law

- L114-115: reference for 50% figure?

- L148: reference for melt due to GHF?

- L537: reference for observed rate

We will add supporting citations for these statements

L106: what is the threshold for the limit on superimposed ice, and where does this come from? How sensitive is runoff to this threshold – couldn't this also feed into the Bayesian inference?

Following the description of Janssens and Huybrechts (2000), we use the capillary retention model that takes into account the refreezing process and the capillary suction effect of the snowpack. The potential-retention fraction is given by:

$$p_r = min\left[\frac{c}{L}T.\frac{C}{P} + \left(\frac{C-M}{P}\right).\left(\frac{\rho_e}{\rho_o} - 1\right); 1\right]$$

Where c and L are the specific heat capacity and latent heat of fusion of ice, P is the mean annual total precipitation, T is the mean annual temperature, M is the total annual snowmelt, $\rho_o$ and $\rho_e$ are the densities of dry snow and water-saturated wet snow, respectively. We will give more details in the revised manuscript.

Regarding the sensitivity of runoff to this threshold; certainly, there will be an affect and therefore a sensitivity that could feed into the Bayesian inference. We selected two parameters of the PDD model to explore two mechanisms: the increase in precipitation and increase in temperature extremes as CESM1 temperatures change during the course of our simulations. The parameter $\sigma_M$ already essentially changes the amount of runoff in our model, and so sensitivity to this parameter and the retention fraction would presumably be similar. Since the difficulty of our Bayesian inference and the number of simulations required scales nonlinearly with the number of parameters we explore, adding another parameter would

only make sense if that allows to include an entirely different mechanism. The degree day factors, for example, could also be included but we did not for the same reason.

L113-114: This sentence implies that there are no other contributors to basal melt of grounded ice – perhaps worth clarifying that this is referring to your model rather than reality. For example, geothermal heat flux is mentioned later on as being of less importance, but this sentence implies it doesn't contribute to melt at all.

We will clarify this and include a comment on geothermal heat flux in the revised manuscript.

Figure 2: caption could include more detail, e.g. define the colormaps. In the main text, are you able to offer any physical interpretation of what these PCs are showing?

We will include a colormap in the revised manuscript. Regarding a physical interpretation of the PCs; broadly speaking PC1 represents an overall thinning and retreat signal of the entire ASE region, which is unsurprisingly the main mode of change in the region, and subsequent PCs represent variation in this response between the Pine Island, Thwaites and Dotson/Crosson catchments. We are cautious about ascribing these smaller PCs to any one particular physical mechanism, however, they are more like ingredients that can be combined in different amounts to yield a spatial pattern that most closely matches the result of each simulation, and the mechanism behind that spatial pattern could be very different in each case.

L369: is the assumption that observational errors are spatially uncorrelated robust? E.g. velocity errors (in magnitude and direction) are dependent on flow speed.

This is a simplifying assumption that makes the probabilistic inference easier, but one that is commonly used in similar papers, e.g. Wernecke (2020)

L446 / Section 4: the interpretation and discussion in this section could be expanded. For example, why do think that the weight given to surface elevation change in the inversion (along with other inversion parameters) is the biggest contributor to uncertainty? Does this indicate that the model is most sensitive to C and A, or are the inversion parameter priors simply considerably wider than those of other parameters? Given the sensitivity, what is the implication therefore of keeping the C and A fields constant throughout century-scale forward simulations (I realise this is the way it's done, but interested if your results give further insight into whether this can really be justified)? How do the various inversion parameters translate to variation in C and A, in magnitude and spatially? Do these fields differ a lot between the 1996 and 2021 initialisation?

We will expand on this section in the revised manuscript, but we address these questions directly. There are a number of things to disentangle here, but essentially what is clear is that the parameters related to our model initialisation contribute the most to uncertainty and among those, the weight given to the surface elevation change in the inversion through parameter $\dot{h}_E$ is the most important.

The model response is indeed sensitive to spatial variability in the A and C fields, as would be expected since for uniform A and C fields the model would completely fail to match observed velocities. However, we don't expect the parameter priors play an important role, since in the Bayesian framework the importance of the prior diminishes rapidly as you increase the amount of 'data', or in our case model simulations. Comparing the A and C fields between simulations is challenging, since every inversion simultaneously changes every parameter including $m$ and $n$, which therefore change the units of A and C. This also precludes a comparison between 1996 and 2021 initialisations, since they are separate experiments sampling different parts of the parameter space so again no two simulations will have the same combination of m and n. Regarding the assumption that A and C fields do not change during the course of our simulations, clearly this is a limitation of many ice sheet modelling studies. We would argue that for forecasts over relatively short timescales (80 years for most of our simulations) this assumption is not necessarily a bad one, however we cannot quantify that with these results and are merely

speculating. Again, we will expand on a discussion of these results and their implications for modelling studies that use this type of initialisation.

For some parameters there is a big difference in the Sobol indices between RCP8.5 and Paris2C (e.g. basal mass balance parameters) – could you comment on this?

When comparing all Sobol indices between the two scenarios there is a general tendency for RCP8.5 to show more sensitivity to parameters related to external forcing (p, Gamma_{TS}, E_{ID}) and in contrast for Paris2C simulations to show more sensitivity to parameters related to internal ice dynamics (m and inversion parameters). This makes sense since, as we show in the paper, the changes in mass balance are the main cause of differences between these two simulations in terms of cumulative sea level contribution. We will add a comment emphasising this point in the revised manuscript.

L478: perhaps worth noting that this rate is very similar to the present day observed rate for the sector.

We will add a note to this effect.

L485-486: "Secondly, using adaptive mesh refinement…" – many of the studies cited above use BISICLES, which also uses AMR, with a resolution of 250 m at the grounding line, so this sentence is a bit misleading. In general, this paragraph could be developed further – e.g. the inclusion of a plume model rather than a very simple melt rate forcing used by some of the others could be discussed.

We agree this should be framed better, since no single aspect of our study is entirely new and the novelty is in bringing all of these components together, so we will reword this in the revised manuscript.

L561-564: It seems strange to introduce the concept of testing the reversibility in the final paragraph of the conclusions. Perhaps it would be better suited to the discussion, along with the relevant citations.

We will add some brief discussion related to this earlier in the revised manuscript.

L580-581: I think limiting ocean driven melt to strictly ungrounded nodes is a sensible and widely used approach, but here or elsewhere, perhaps it is worth commenting on recent modelling and observational work that indicates that ocean water intrusions far upstream of the grounding "line" could be contributing significantly to melt? E.g. Bradley and Hewitt (2024, 10.1038/s41561-024-01465-7), Rignot et al. (2024, 10.1073/pnas.2404766121).

We will add a remark along these lines in the revised manuscript.

Appendix B: perhaps I missed this, but how do you sample the parameter space for the Ua-obs ensemble used to train the surrogate?

Parameters are sampled using latin hypercube sampling and we will clarify this in the revised manuscript.

**Technical corrections**

Please ensure that all abbreviations/acronyms are defined at point of first use (e.g. SLR, line 22; RNN, figure 1).

Table 1: check inversion parameter symbols vs names

Check formatting of citations, e.g. years not in brackets (e.g. L151), use of et al. vs and others.

L348: thinnning -> thinning

L390: remove extra "model"

Fig. 4: E_dhdt -> hdot_e?

Fig. 7: hdot_E -> hdot_e (probably worth checking consistency of symbols throughout)

L440: colormap for GLs is brown (as indicated by the figure 6 caption) not red.

L585: "is draft" -> "ice draft"?

All technical corrections will be implemented in the revised manuscript.

**References:**

Janssens I, Huybrechts P. (2000) The treatment of meltwater retention in mass-balance parameterizations of the Greenland ice sheet. *Annals of Glaciology* **31**:133-140. doi:10.3189/172756400781819941

Wernecke, A., Edwards, T. L., Nias, I. J., Holden, P. B., and Edwards, N. R. (2020) Spatial probabilistic calibration of a high-resolution Amundsen Sea Embayment ice sheet model with satellite altimeter data, The Cryosphere, **14**, 1459–1474, https://doi.org/10.5194/tc-14-1459-2020.

---

## Author Response (AR1)

We thank the reviewers for their time and effort in providing detailed feedback on our manuscript. Our responses to their comments are given below, in red, and the changes we have made to the manuscript in response to these changes are in bold green.

Comments from Reviewer 1

**General comments**

This paper presents a thorough and rigorous method for producing calibrated projections of sea level rise from the Amundsen Sea sector of West Antarctica, using a multistep process to consider and constrain uncertainty from a wide range of sources, by combining both a numerical ice flow model and a statistical surrogate model. The resulting sea level contributions by 2100 under RCP8.5 and Paris2C are at the lower end of the range of estimates from previous studies and similar between the two scenarios. They also extended some simulations to 2250 and found the two scenarios diverged after 2100 due to differences in snow accumulation.

In general, the study is very well presented (save a few consistency errors – see technical corrections below) and structured clearly, with a lot of methodological detail. However, I think certain later sections could benefit from further discussion and interpretation, e.g. Section 4, in particular, is very brief and offers no real interpretation, which is a bit of a shame – see specific comments for more detail.

**We have added extensive additional discussion and interpretation of our results, as described in more detail in response to specific comments, below.**

**Specific comments**

There are several places where statements are made without supporting citations:

- L83-84: give examples of studies that use a mixed sliding law **Done**

- L114-115: reference for 50% figure? **Done**

- L148: reference for melt due to GHF? **Done**

- L537: reference for observed rate **Done**

L106: what is the threshold for the limit on superimposed ice, and where does this come from? How sensitive is runoff to this threshold – couldn't this also feed into the Bayesian inference?

Following the description of Janssens and Huybrechts (2000), we use the capillary retention model that takes into account the refreezing process and the capillary suction effect of the snowpack. The potential-retention fraction is given by:

$$p_r = min\left[\frac{c}{L}T.\frac{C}{P} + \left(\frac{C-M}{P}\right).\left(\frac{\rho_e}{\rho_o} - 1\right); 1\right]$$

Where c and L are the specific heat capacity and latent heat of fusion of ice, P is the mean annual total precipitation, T is the mean annual temperature, M is the total annual snowmelt, $\rho_o$ and $\rho_e$ are the densities of dry snow and water-saturated wet snow, respectively. We will give more details in the revised manuscript.

Regarding the sensitivity of runoff to this threshold; certainly, there will be an affect and therefore a sensitivity that could feed into the Bayesian inference. We selected two parameters of the PDD model to explore two mechanisms: the increase in precipitation and increase in temperature extremes as CESM1 temperatures change during the course of our simulations. The parameter $\sigma_M$ already essentially changes the amount of runoff in our model, and so sensitivity to this parameter and the retention fraction would presumably be similar. Since the difficulty of our Bayesian inference and the number of simulations

required scales nonlinearly with the number of parameters we explore, adding another parameter would only make sense if that allows to include an entirely different mechanism. The degree day factors, for example, could also be included but we did not for the same reason.

**We have added more details on the capillary retention model in the description of the surface mass balance model.**

L113-114: This sentence implies that there are no other contributors to basal melt of grounded ice – perhaps worth clarifying that this is referring to your model rather than reality. For example, geothermal heat flux is mentioned later on as being of less importance, but this sentence implies it doesn't contribute to melt at all.

**We have clarified that this refers only to our model and given an expanded explanation of why we do not include the geothermal heat flux term.**

Figure 2: caption could include more detail, e.g. define the colormaps. In the main text, are you able to offer any physical interpretation of what these PCs are showing?

We will include a colormap in the revised manuscript. Regarding a physical interpretation of the PCs; broadly speaking PC1 represents an overall thinning and retreat signal of the entire ASE region, which is unsurprisingly the main mode of change in the region, and subsequent PCs represent variation in this response between the Pine Island, Thwaites and Dotson/Crosson catchments. We are cautious about ascribing these smaller PCs to any one particular physical mechanism, however, they are more like ingredients that can be combined in different amounts to yield a spatial pattern that most closely matches the result of each simulation, and the mechanism behind that spatial pattern could be very different in each case.

**We have added an expanded figure caption and colorbar along with some discussion of possible interpretations for the principal components plotted in the Figure.**

L369: is the assumption that observational errors are spatially uncorrelated robust? E.g. velocity errors (in magnitude and direction) are dependent on flow speed.

This is a simplifying assumption that makes the probabilistic inference easier, but one that is commonly used in similar papers, e.g. Wernecke (2020)

L446 / Section 4: the interpretation and discussion in this section could be expanded. For example, why do think that the weight given to surface elevation change in the inversion (along with other inversion parameters) is the biggest contributor to uncertainty? Does this indicate that the model is most sensitive to C and A, or are the inversion parameter priors simply considerably wider than those of other parameters? Given the sensitivity, what is the implication therefore of keeping the C and A fields constant throughout century-scale forward simulations (I realise this is the way it's done, but interested if your results give further insight into whether this can really be justified)? How do the various inversion parameters translate to variation in C and A, in magnitude and spatially? Do these fields differ a lot between the 1996 and 2021 initialisation?

We will expand on this section in the revised manuscript, but we address these questions directly. There are a number of things to disentangle here, but essentially what is clear is that the parameters related to our model initialisation contribute the most to uncertainty and among those, the weight given to the surface elevation change in the inversion through parameter $\dot{h_E}$ is the most important.

The model response is indeed sensitive to spatial variability in the A and C fields, as would be expected since for uniform A and C fields the model would completely fail to match observed velocities. However, we don't expect the parameter priors play an important role, since in the Bayesian framework the importance of the prior diminishes rapidly as you increase the amount of 'data', or in our case model simulations. Comparing the A and C fields between simulations is challenging, since every inversion simultaneously changes every parameter including $m$ and $n$, which therefore change the units of A and C.

This also precludes a comparison between 1996 and 2021 initialisations, since they are separate experiments sampling different parts of the parameter space so again no two simulations will have the same combination of m and n. Regarding the assumption that A and C fields do not change during the course of our simulations, clearly this is a limitation of many ice sheet modelling studies. We would argue that for forecasts over relatively short timescales (80 years for most of our simulations) this assumption is not necessarily a bad one, however we cannot quantify that with these results and are merely speculating. Again, we will expand on a discussion of these results and their implications for modelling studies that use this type of initialisation.

**We have added an additional paragraph discussing the implication of our modelled sensitivity to parameters related to the inversion.**

For some parameters there is a big difference in the Sobol indices between RCP8.5 and Paris2C (e.g. basal mass balance parameters) – could you comment on this?

When comparing all Sobol indices between the two scenarios there is a general tendency for RCP8.5 to show more sensitivity to parameters related to external forcing (p, Gamma_TS, E_ID) and in contrast for Paris2C simulations to show more sensitivity to parameters related to internal ice dynamics (m and inversion parameters). This makes sense since, as we show in the paper, the changes in mass balance are the main cause of differences between these two simulations in terms of cumulative sea level contribution. We will add a comment emphasising this point in the revised manuscript.

**We have added an expanded interpretation on the differences in Sobol indices between RCP8.5 and Paris2C.**

L478: perhaps worth noting that this rate is very similar to the present day observed rate for the sector.

**We have added a comment noting this similarity and a citation to Davison et al. (2023)**

L485-486: "Secondly, using adaptive mesh refinement..." – many of the studies cited above use BISICLES, which also uses AMR, with a resolution of 250 m at the grounding line, so this sentence is a bit misleading. In general, this paragraph could be developed further – e.g. the inclusion of a plume model rather than a very simple melt rate forcing used by some of the others could be discussed.

**We have added an expanded discussion on how our study differs from other similar studies, including the surface and basal mass balance parameterisations, and now make it clear that our mesh resolution at the grounding line is not necessarily higher than all aforementioned studies.**

L561-564: It seems strange to introduce the concept of testing the reversibility in the final paragraph of the conclusions. Perhaps it would be better suited to the discussion, along with the relevant citations.

**We have retained this sentence in the conclusion but now add an expanded discussion on this point at the end of the results section**

L580-581: I think limiting ocean driven melt to strictly ungrounded nodes is a sensible and widely used approach, but here or elsewhere, perhaps it is worth commenting on recent modelling and observational work that indicates that ocean water intrusions far upstream of the grounding "line" could be contributing significantly to melt? E.g. Bradley and Hewitt (2024, 10.1038/s41561-024-01465-7), Rignot et al. (2024, 10.1073/pnas.2404766121).

**We have added a comment on this caveat along with the suggested citations**

Appendix B: perhaps I missed this, but how do you sample the parameter space for the Ua-obs ensemble used to train the surrogate?

**Parameters are sampled using latin hypercube sampling and this is explained at the end of Sect. 2.4.3**

**Technical corrections**

Please ensure that all abbreviations/acronyms are defined at point of first use (e.g. SLR, line 22; RNN, figure 1). **Done**

Table 1: check inversion parameter symbols vs names **Fixed a mistake in the row order**

Check formatting of citations, e.g. years not in brackets (e.g. L151), use of et al. vs and others. **All citation formatting issues should now be fixed**

L348: thinnning -> thinning **Done**

L390: remove extra "model" **Done**

Fig. 4: E_dhdt -> hdot_e? **Done**

Fig. 7: hdot_E -> hdot_e (probably worth checking consistency of symbols throughout) **Done**

L440: colormap for GLs is brown (as indicated by the figure 6 caption) not red. **Done**

L585: "is draft" -> "ice draft"? **Done**

Comments from Reviewer 2

**General Comments:**

In this discussion, the authors introduce a new methodology for comprehensively determining uncertainty of ice sheet model projections. The study focuses on modeling ice sheet change in the Amundsen Sea sector of Antarctica under two different future emission pathways through 2100 and then through 2250, using the ice sheet model Úa. The authors design a method to derive uncertainty in their projections by training a surrogate model and then using Bayesian calibration to down select the most reasonable parameter space for their set of historical simulations. They then sample the calibrated parameter distributions, running an ensemble of future projections, which are in turn used to train a new, time-dependent surrogate model to exhaustively cover the possible parameter space. Results allow the authors to quantify uncertainty for every year of their projection and attribute uncertainty to each of their various parameters. They conclude that uncertainty is dominated by parameters related to initialization of basal sliding and ice rigidity for both scenarios followed by parameters related to ice flow and basal melting. Finally, the authors extend a subset of simulations in time through 2250 and find that mass loss accelerates through the year 2200, returning to a sea-level contribution similar to today's by the end of the simulation, for both scenarios similarly. Results are compared against projections from previous Bayesian-based studies, concluding that the sea-level projections for this study are more conservative than others, especially considering that the simulations presented include a MICI mechanism for ice shelf collapse.

This manuscript is well written and of high quality. The work is novel, and the figures are well designed, easy to read, and support the stated conclusions. The experiments themselves required a significant amount of work and thoughtful effort toward their design. The authors take an organized approach to describing the complex workflow, including a helpful schematic figure and extensive appendices. For these reasons, I support the publication of this manuscript in The Cryosphere.

However, from a scientific point of view, I think the authors could expand upon their discussion of the results more. I realize that the main point of the paper is to introduce the novel workflow to quantify uncertainty in ice sheet model projections, but the authors do make a point to compare their projection results against other published projections for the region. They also conclude that their projections are more conservative than others, showing very little signs of instability (even though MICI processes were invoked during the simulations). Because of this, I believe that the authors should do more than just

show the end states of their projection ensembles; it would greatly enhance the manuscript if they also discussed why the simulations are tracking on the low end of sea-level contribution. This includes expanding the discussion to investigate why the simulations do not match present-day sea-level contribution until 2200 and why the simulations for both scenarios increase so distinctly in the extended simulations (at year 2100). I also think it would improve the manuscript if the authors noted some implications for their stated results and conclusions. Finally, I find that in some cases, the method description lacks detail and is vaguely described; it would aid the reader to have them clarified and/or quantified before publication. More specific comments are noted below.

**We have expanded on the interpretation and discussion of our model results, as requested by all reviewers. We intentionally avoided adding too much information into the methods section since this is already very long, but where the reviewers have requested more details as noted in specific comments below, these have been added.**

**Specific comments:**

Fig. 1: From my understanding, both the historical runs and the forward runs simulate the year 2021. Is that correct? If not, please clarify this throughout the text.

The Ua$_{obs}$ simulations cover the period 1$^{st}$ Jan 1996 to 1$^{st}$ Jan 2021, whereas the Ua$_{fwd}$ simulations begin on the 1$^{st}$ January 2021, so there is no overlap and this will be clarified in the text.

**This is now state very clearly in the text**

Table 1: I suggest that you also include the ranges for these parameters in the table. That would be very handy information for the reader to have for reference.

We agree that this would be helpful, although using the range only makes sense for parameters with a uniform distribution and is not easy to interpret for parameters with a Gaussian probability distribution, so we will consider how best to incorporate this information without confusing the reader.

**We now show the range for prior parameters in the table, where range is the minimum and maximum for parameters with uniform priors and the 3\sigma range for parameters with Gaussian priors, as explained in the table caption.**

Lines 78-79: ... along "with" other sliding laws. Additionally, please quantify "some time". It would also be helpful to the reader if other key sliding laws were explicitly listed (i.e. Coulomb and Weertman).

**Done**

Line 94: Since you discuss the bias-corrected aspect later in the paper, it would be helpful if you noted that the correction later described (and perhaps reference the section) for the reader.

**Done**

Line 106: Please include what this threshold is in the text.

**We use the capillary retention model as described in Janssens and Huybrechts (2000) and we have added a brief description of this in the revised manuscript.**

Line 148: "which is at least two orders of magnitude larger than basal melting due to geothermal heat flux." Is there a reference that can be added here for a noted estimate of geothermal heat flux magnitude?

**We have expanded on this and added a reference for melting due to geothermal heat flux in the region.**

Line 210: Please quantify what qualifies as "large" spatial gradients.

**In fact the regularisation penalises any spatial gradient in the inverted field, but larger spatial gradients are penalised more. We have removed the word "large" here to make this less ambiguous.**

Line 216: Please specify mass balance, surface+basal mass balance?

**We have clarified this means surface and basal mass balance.**

Line 219: "after which all model forcing terms evolve based on the ERA5 outputs for the period 1996-2021." This statement is somewhat confusing, because if I understand correctly, the ocean forcing does not rely on ERA5. Similarly, my understanding is that the other (atmospheric) historical forcing is not dictated by ERA5 either (but corrected CESM1?). Please rephrase this section for clarity. Also, ERA5 or the historical forcings have not yet been introduced at this point in the paper, so it is unclear what you mean by "ERA5 outputs". Please either introduce (reference) it here or make a reference to Section 2.2 to help guide the reader.

**Thanks to the reviewer for spotting this mistake, line 219 should refer to CESM1, not ERA5, and this has now been corrected.**

Line 224: Please give a value for the initial coarse resolution.

**The coarser resolution mesh is itself unstructured and so it does not have a specific resolution.**

Lines 239-241: This statement is confusing to the reader because RCP4.5 is noted as running through 2080, but the obs simulations only run through 2021, is that correct? Also it not clear what "that most closely matches observed atmospheric conditions in the model domain" means. Please rephrase.

**Both of these points are now clarified in the manuscript**

Line 243: Please add a reference for the CESM1 simulations used here, as well as the ensemble simulations. This part of your method discussing the inclusion of climate variability is quite vaguely described.

**We have slightly expanded the description here and included references for the CESM1 model and simulations.**

Line 252: Is the bias correction done monthly (for temp and precip)? Is the temperature correction spatially varying as well? Please add a more precise description of this method.

Bias correction in both cases is done with a spatially varying but constant in time term, so we assume that CESM1 has a systematic bias, whose severity depends on location, but which does not change as a function of e.g. present climate.

**We have explained this more clearly in the revised manuscript**

Lines 257-258: Please clarify if the bias correction is done on all precipitation and temperature forcing, even the fwd simulations. In this case, if the historical bias correction for precipitation was used on the future forcing, then the larger precipitation values grow, the larger the correction (because a scaling factor is applied)? In this case, couldn't the surface mass balance future trend be altered by the precipitation scaling? Could you comment on whether or not that is a concern?

The bias correction is indeed applied in the same way for all simulations, however the term 'scaling factor' was poorly chosen, we use the same approach as Naughten et al. (2022) whereby a spatially varying field calculated as the difference between time averaged CESM1 precipitation/temperature and our 'preferred' dataset e.g. AntAWS/ERA5/MAR over the same period. This difference is considered to be the bias and the spatially varying bias field is added to the CESM1 field, so the size of the bias term will not be affected by changes in the temperature/precipitation fields with time.

**This explanation is now expanded upon in the revised manuscript**

Line 265: Please specify how this is done. Point by point on the forcing grid? Is it for temperature only, or also for the ocean modeling? I am curious about whether this affects your trend when you restart your simulations, as it is not clear why both emissions begin increasing in sea level contribution starting right

at 2100 (Figure 9).  The RCP8.5 in particular changes from a downward trend to an upward trend almost instantaneously.  Have you diagnosed what causes the sudden retread based on your simulation ensemble?  This is an example of a curious model response that could be explained and discussed in more detail in the paper.

For every point on the forcing grid, for both temperature, precipitation, and all MITgcm model forcing, a linear trend is calculated for the period 2080-2100. The forcing for all fields for the years 2100-2250 is then taken as a repeat of this final period with the calculated linear trend removed. An example time series of temperature from a point in the domain is given in the figure below (gray line is monthly temperature, yellow line is moving average temperature with a 12 month window).

[Figure]

Regarding what the reviewer highlights as sudden retreat in Fig. 9, firstly this figure shows sea level contribution rate in mm/yr, so it is not directly related to retreat and the change in the (average) rate that appears to happen at around the year 2100 would therefore be much less apparent if plotted as change in ice volume.  That being said, there is a clear change in the trend, which is stronger for RCP8.5 than Paris2C. Attributing this change which is an average across all simulations with grounding lines in different positions is challenging but the most likely cause is simply as a result of changes in surface mass balance, since the trend of increased precipitation and warming is removed after 2100, this also explains why the change is more notable for RCP8.5 where these changes are stronger. As mentioned elsewhere, we are re-running a representative sample of simulations to enable a more thorough analysis of model behaviour and this change in trend will be one focus of that analysis.

**We have added a clearer description of the detrending that is described in this section. We have also significantly expanded our discussion of model results including the change in mass loss rate that occurs at around the year 2100.**

Lines 276-277: "Thus, the simulation start dates of 1996 and 2021 were chosen to ensure that the velocity and surface elevation change data were approximately aligned in time, but the precise timing is not well defined."  This sentence is awkward, and I am not sure what it means, please rephrase.

What we mean is that these dates were chosen to align as best as possible with all datasets that were used, but the actual timings of these datasets are (a) not precisely defined anyway and (b) do not match

one another exactly, and so these chosen dates are a compromise that tries to minimise discrepancy in the acquisition time of each dataset and keep things as consistent as possible.

**We have rephrased this sentence in the revised manuscript.**

Lines 311-312 and Fig. 2/3: My understanding is that you are focused on change of surface elevation and change in surface speed during your obs period. However, I have found that the wording to describe these terms throughout the manuscript is confusing. Sometimes you refer to just surface speed (i.e. Fig 2 is labeled surface elevation and velocity – a side note, in this case, perhaps velocity should be changed to speed?). The Fig 2 caption also specifies that you are plotting variable change: i.e., "surface ice speed (top row) and surface elevation change (bottom row)", but this sounds like you are considering only change with respect to the surface elevation, not surface speed. This is similarly confusing in Fig. 3, since the plot labels include a delta, but the caption is not clear that it is delta speed being shown. In general, speed seems to be referenced in many sections of the paper, but I do not find it explicitly stated in each instance that the benchmark of interest is "change" in speed. This just might be a question of wording and can be fixed easily. Please try to make this clear and consistent all throughout the manuscript.

This is indeed a question of poor wording. In all cases our 'observations' that we use to calculate model parameter likelihoods are change in surface elevation and change in surface ice speed. We will carefully go through the manuscript and ensure this is corrected or clarified in all cases.

**We have corrected the wording in figures 2 and 3 and been through the manuscript to make sure this is consistent throughout.**

Line 326: Please note what the resulting k values are in the main text for this analysis and Fig. 3.

**We have added this information to the main text (note k=8 for surface elevation change and k=11 for change in surface ice speed).**

Fig. 2: In my understanding, your calibration is based on the change of speed and the change of thickness over the observational period of interest. Do you think there are implications to choosing these diagnostics and how do you think it affects your results? For instance, you are testing for a linear change but change in this area is temporally variable and non-linear. Even the clear strong trend captured by the PC1's (responsible for a significant portion of the changes) are in reality much more complex temporally. What are the repercussions of this assumption? Mentioning these caveats in the manuscript and discussing on how these constraints were decided upon (or computationally forced) would be a very interesting addition to the paper. This is especially the case because difficult decisions and concessions like these are likely why many others have not attempted an assessment of this magnitude.

This is a very valid point and we agree that it would be beneficial to add a discussion of this to the revised manuscript. While the data exists to extend our calibration to include temporal variability in the regional response, and this could help further constrain the model, this approach may currently be too ambitious in terms of how well the current generation of models can represent processes relevant over shorter timescales (e.g. calving, damage, hydrology, ocean processes etc.). While studies do exist that seek to match observed temporal variability over shorter timescales, they are generally focused on one outlet glacier or limited in other ways. Matching the differing response of each portion of the Amundsen Sea Sector over the entire satellite record would be a significant challenge without some modelling compromises such as enforcing the observed calving front positions rather than allowing them to evolve dynamically as we do in this study. We also arguably still lack sufficiently detailed knowledge of bedrock geometry that could greatly impact retreat rates over shorter timescales. That being said, the assumption we are making by only comparing speed and surface elevation at the start and end of each simulation is an important one, and extending this study to include more data for calibration would be an interesting and potentially valuable addition for future work.

**We have added several sentences of this simplifying assumption later in the discussion**

Lines 347-348: With respect to capturing the trend and magnitude of these constraints spatially, do you think that PC2+ play an important factor in capturing the variability despite the lower dimensionality? Having done all the work to assess the presented method, how reasonable do you think this method of calibration is, considering mass loss in this area is so non-linear?

As noted in the paper, using just PC1 would only account for 64% of variability in ice speed change and 57% of variability in surface elevation change, so additional terms are certainly needed to help capture the complex response of this region. We refer to our answer above regarding non-linearity.

Lines 359-360: Please note or reference these physically plausible values, or reference where it is discussed (e.g. Appendix C).

**We have added a reference to appendix C here.**

Line 413: Quantify "large".

**Done**

Fig 4: As you mention in Appendix B3, for instance, values for n have been questioned recently. Your calibrated ranges for m and n are interesting to see, and they are a nice result in themselves. It would be great to see some discussion on this result added to the text.

It is indeed interesting that for our calibration we find an optimal value for n centred around 4, rather than 3 that is more commonly used.

**We have added several sentences on this in the revised manuscript.**

Line 424: Instead of using "~", please specify the exact number of simulations. Also, please note here that it is 2000 simulations in total (not for each of the scenarios as I understand it).

**Done**

Fig 5: This is a really nice figure, with a significant amount of information portrayed. While it is clear that the sea-level projections are conservative, and this is amply noted in the manuscript, it would improve the text if you discussed the reasoning for why the trend of sea level contribution is lower than the past observed long-term trend, and much lower than the present-day trend. I would be curious to know what is happening dynamically to slow down the sea-level contribution, and an interpretation of what those results mean (i.e., line 532 suggests that by 2100 the rate of ice loss is ½ of the present-day rate, which is a quite surprising behavior – what is stabilizing the model projections?). For instance, it might be that the majority of your simulations are generally stuck on their current grounding line (but it is surprising that it would happen to be for all of them), or there is something about the initialization constraints that do not allow the transient fwd simulations to contribute as much sea-level as is observed at their start (present-day). Whatever is the cause, an analysis of the simulations could reveal a general conservative trend that many of ensemble members (ice sheet model runs) seem to be following.

We concede that the paper would benefit from more analysis to better understand why the model is behaving as it is, although attributing this behaviour to one thing for a complex model with many interacting components and thousands of simulations is presumably not going to be possible. As mentioned elsewhere, we could not save most model output fields for the thousands of simulations due to storage constraints, making more detailed analysis challenging. We are in the process of re-running a representative sample of simulations with detailed outputs and we will explore in more detail the reasons why the model behaves as it does using this sample for the revised manuscript.

**We have included extensive additional analysis and discussion exploring these points and other similar points raised by the reviewers, both expanding on our comparison with previously published work and delving deeper into why our simulations behave as they do.**

Line 455: It would be interesting for you to comment in the paper if you think this happens because of the type of inversion method being used for this study? That is, there are larger degrees of freedom than if you just inverted for one parameter? Do you think this uncertainty would be as strong if only basal sliding was inverted for, for instance? Could you make a statement in the discussion about the implication for this and choices for model initialization impacting not only future projections but their uncertainty?

This is an interesting question, although we can only speculate on the implications of this choice for uncertainty. The Ua ice flow model inverts for both basal slipperiness and ice rate factor everywhere in the domain because, in the absence of far more detailed observations and a modelling framework to capture complex processes such as damage, hydrology etc. this is the only way to ensure that the ice sheet initial conditions (e.g. surface velocities) match what is observed. If we were to only invert for basal slipperiness, the model would not match observations as well, and in this respect uncertainty would be higher.

**We have added an expanded discussion on the sensitivity to initialization parameters in the revised manuscript**

Line 494: The Sun et al., 2014 is based on Bedmap2 uncertainties, perhaps you could add some additional references for more recent papers that support this hypothesis using BedMachine or similar.

**We have added references to Wernecke (2022) and Castleman (2022) which are more recent examples that include Bedmachine.**

Lines 536-537: As discussed above, please add some assessment of why this is the case and what the implications are for these results, including the sudden increase in sea-level contribution in the extended projections. In addition, do you know what causes what looks like another stabilization (and even possible downtrend in sea-level contribution in Paris 2C) around year 2200? Fig. 6 is very helpful to see that grounding lines do significantly retreat during the extended simulations, but have you seen any reasoning for why when analyzing the results?

As mentioned above, once a sample of simulations have been repeated with additional outputs this will greatly enhance our ability to dig deeper into the behaviour of our simulation ensemble and we will add more analysis related to this and other points raised elsewhere in the revised manuscript.

**We include new plots, analysis and discussion to address these questions and other similar ones in much more detail.**

Line 541: Please make a statement about what it means that these two scenarios are so similar. (In some ways this does make sense with the attribution analysis presented, that model setup and initialization is more important than forcing). Does this mean that ice sheet modelers should take particular caution in initializing their models, especially as computational ability allows for a more complex method and therefore possibly more unknows?

Certainly, initialisation plays a key role in the model response to any forcing, as demonstrated in our analysis. In this region in particular, much of the observed grounding line retreat and acceleration of grounded ice may be a response to past perturbations that the glaciers are still adjusting to. Attempting to replicate these signals without necessarily including the original forcing that triggered them is challenging and although our inversion includes observed thinning rates which helps ensure that the ice sheet is initialised on the rate trajectory, the retreat history cannot be fully accounted for without having detailed observations going back further in the past.

**We have added more discussion on this point, both here and in the section on sources of uncertainty.**

Fig 9: Could you add a line where the current (or historical) sea-level rate of change is for comparison?

**Done**

Line 552: Could you make a statement about the implication of these results?

**We have added a paragraph discussing this result and its implications.**

Line 553: Do you think that your historical being linear and calving/climate variability being partly stochastic drives this result in any way?

This is an interesting thought, and maybe partly explains the result. Regarding calving, while this may be stochastic in reality, the calving law that we use (with reasoning behind our choice explained in the paper) is not stochastic and in fact behaves very similarly no matter what else is happening within the model (as can be seen to some extent in Fig. 6). There would be substantial value in exploring uncertainty related to different calving laws, although arguably no calving law currently exists which can be easily implemented and captures the complex and stochastic nature of calving. In terms of internal climate variability, presumably a longer calibration period would increase the importance of this term significantly and 25 years is not really sufficient to fully capture the most interesting decadal variability observed in the region.

Fig E1: This is a nice, and very helpful, plot to include. It suggests that the pdf for your simulations should not be that different from what the surrogate model outputs. Is that correct? For the values of this plot, I think this is accumulated sea-level contribution at the end of each year. Please clarify what the values of the axes are in the caption, to prevent confusion.

The figure aims to demonstrate the fidelity of the LSTM surrogate to the Ua-fwd simulations and so by the same token they should have similar pdfs, yes. The reviewer is correct, the figure shows cumulative sea level contribution and this will be clarified in the revised manuscript.

**We have now made it clear in the caption that these are cumulative sea level contributions.**

**Technical corrections:**

Fig. 1: Please define RNN, GMSL, and LSTM for this figure. Some are defined in other parts of the text, but it would be helpful for the reader to have it spelled out here too. **Done**

Line 83: "popular" => I suggest using more formal wording here, like "used more frequently by the community", or something similar that invokes a scientific backing. **Done**

Lines 89-91: "Note that other contributions to changes in precipitation (and hence accumulation) ..." This sentence is difficult to understand. Please rephrase. **Done**

Line 100: Please make sure symbols are clearly defined (i.e. $\sigma_M$, $T_{cesm}$) **Done**

Line 154: Please make sure symbols are defined (i.e. $r$, $r_c$) **Done**

Line 200: Bedmachine => BedMachine **Done**

Line 234: "this study" sounds like your study, but I think you mean the Naughten study? Please specify. **Done**

Line 330: "changed" => change **Done**

Fig. 3: Caption should have elevation as the top row and speed as the bottom row. **Done**

Fig 7: On the x axis: $h_E$ => $h_e$ **Done**

Appendix A2: Please update the title so it reflects that this is specifically for ocean-induced melt **Done**

Line 742: "an Long" => "a Long" **Done**

Line 752: This statement does not need to be approximate. Please include the exact number of simulations used. **Done**

Comments from Reviewer 3

**Overview**:

In this research article, Roiser et al. present numerical ice sheet modeling results of the Amundsen Sea Embayment (ASE), a dynamic region of West Antarctica that is rapidly contributing to global sea level, to 2100 and 2250 under RCP8.5 and Paris-2C emission scenarios. Central to the paper is the development of a new framework for quantifying uncertainties associated with these modeling results by training a surrogate model. Overall, the authors find that the sea level contribution of the ASE through 2100 is nearly identical in both RCP8.5 and Paris-2C, tending to the lower range of other published sea level projections of this region. Grounding line migration is minimal through 2100; however, iceberg calving drives near-total loss of all floating ice shelves. Beyond 2100, the simulations diverge as snowfall ramps up in RCP8.5. In terms of the uncertainty quantification, the authors found that parameters related to model initialization (e.g., hE and the inversion coefficients) comprised a majority of the uncertainty, followed by ice flow parameters, basal mass balance parameters, and surface mass balance parameters.

Overall, I find this manuscript to be of very high quality and I expect the results will be of wide interest to the glaciological and cryosphere community. I also believe that the methods of uncertainty quantification are exceptional and provide critical insight into ice sheet modeling parameters. I do share some similar concerns with the other reviewers in that I would like to see further interpretation of the forward simulation results. In particular, I find it very interesting that there is minimal grounding line retreat but near-complete loss of floating ice shelves through 2100 – I would like to see the authors dig into the reasons for this as well as into the reason why their estimates are on the lower end of published sea level estimates of this region. I also have a note in the general comments section that I think it would be appropriate for at least a subset of the ice sheet modeling results to be made publicly available given the broad interest in this region. Lastly, I also have a couple more minor line comments that should be easily addressed. Once these issues are addressed, I would be very happy to support prompt publication of this work in The-Cryosphere.

**All reviewers agreed that the paper would be improved by further discussion and interpretation of model results and we have added this to the revised manuscript. Regarding the lack of grounding line retreat despite loss of ice shelves specifically, as we point out in the discussion this lack of sensitivity to ice shelf loss has already been shown for Thwaites glacier by Gudmundsson et al. (2023), and loss of floating ice shelf in front of Pine Island Glacier is less dramatic than Thwaites in our simulations, so this result is perhaps not entirely surprising. Following the reviewers request, we have also uploaded a subset of model results to a public repository.**

**General Comments:**

**Additional discussion related to modeling results**: Like the other reviewers highlighted, I think that the results of this paper can be bolstered by diving deeper into analysis of the Ua_fwd projections at both 2100 and 2250. In particular, the authors present a unique implementation of the buoyant plume parameterization and a new way of prescribing oceanic temperature and salinity inputs to this parameterization (using the modeled depth of the thermocline as a depth-cutoff in applying T/S). Given that this region of Antarctica is primarily forced by the ocean, I would like to see how these modeled melt rates compare to contemporary estimates and how they change in time with evolving cavity geometry. I also think that comparisons of how much ice mass is added by atmospheric processes versus how much ice mass is advected across the grounding line in the different simulations would be helpful to diagnose why mass loss is generally low in these simulations. Lastly, as I mentioned below in the line comments, I would like to see the introduction of the manuscript expanded to provide an overview of ASE modeling efforts.

**We have undertaken additional simulations and analysis to address these questions, including new figures related to ocean melting and timeseries of the various mass balance terms, and substantial additional discussion of these results, as detailed in response to specific questions below.**

**Some consistent formatting fixes**: Throughout the manuscript, I noticed that in-text citations were not formatted correctly (e.g., in-text citations should be Lazeroms et al. (2018) and not Lazeroms et al. 2018). Also, there are many times in the manuscript where there needs to be a space between the value and the unit (e.g., 1mm should be 1 mm).

**We apologise for the inconsistencies and mistakes that were missed in our initial submission, we have carefully gone through the manuscript and hopefully all these formatting issues are now resolved.**

**Data Availability**: While the authors state that no new datasets were used in this article, they did generate ice sheet numerical modeling outputs that would be of great interest to the cryosphere community given how active numerical modeling efforts are in this area. While I understand that depositing all of the data might be challenging since there are so many model runs, I would implore the authors to deposit a representative subset of the results (or at least the data needed to replicate the figures) in a publicly curated data repository.

We will upload model results to a public repository. We are limited (a) by the way in which model outputs were saved for our two large ensembles, since most fields were only saved at the beginning and end of each simulation and (b) by the size of this dataset, which consists of thousands of simulations. However, we will upload a representative sample that can hopefully be useful to the community.

**We have uploaded all main model fields saved every 2 years for our 60 extended simulations running from the year 2021 to 2300, along with the finite element mesh at each respective time increment. These can now be found in the following public repository: 10.5281/zenodo.14712131**

Line Comments:

Abstract: Much of the paper focuses on the development and implementation of the surrogate model and Bayesian calibration, so I was surprised to see that there was no mention of this in the abstract. If there is room, I would add a sentence about this.
**We have added remarks to this effect in the abstract**

L17: Capitalize "Pine Island" **Done**
L20: Change "rate" to "rates" **Done**

L27: What do you mean by " . . . the complex response of the ocean and ice shelf cavities to atmospheric forcing"? Is this referring to changes in wind stress that impact ocean circulation? For the ice shelves, does this mean enhanced surface melt that might lead to hydro-fracture? Are there studies that you can highlight that show the connection between atmospheric circulation and sub-ice shelf ocean cavity circulation for ASE glaciers? It would be good to specify and connect this to the next sentence because it feels a little ambiguous and out of place currently.
**This is expanded on in the next sentence with citations but that connection is now made clearer after rewording**

L37: Please add a citation for the plume model. **Done**

L15:42: I was hoping to get a little more background on ASE glaciers and recent ice-ocean model studies of this region. I think this will be particularly important if you choose to include additional discussion regarding the results of your ASE sea level rise projections. For example, this would be a great place to discuss what sea level projections of this region exist, what are their shortcomings, what are the partitions of sea level contribution between Thwaites and Pine Island, what ocean and atmospheric forcing datasets and parameterization have been used, have tipping points been found in previous

studies (note that these are examples and you can select what you think would be appropriate)? A bit more background information to help put your research in the context of what has been done already would be very useful.

We agree with the reviewer that a background on ASE glaciers and putting our results in the context of existing studies is important. Currently this is done quite extensively in Sect. 5 of the paper, and we prefer to keep the bulk of this here to avoid repeating things, but we will add more background on ASE glaciers to the introduction following some of the suggested points above.

**We have added an additional paragraph in the introduction that introduces some of these important points at this earlier stage, but hopefully does not repeat too much of the comparison that comes later in the manuscript.**

L44: Add a comma after "uncertainties" **Done**
L45: Change comma to a period after "together" **Done**

L62: What was the minimum and maximum mesh resolution?
**We have added this information to the section dedicated to describing the mesh in Sect. 2.1.7.**

L65: Please add a citation for Glen's flow law. **Done**
L72: Please add a citation for the friction law. Also, what does $m$ control in this friction law?
**A citation exists just below this section and we have added an explanation for the $m$ term in the sliding law.**

L100: Where did the value of 1.66 come from?
**This is a result from the Wake and Marshall (2015) reference given in the next sentence.**

L127: ". . . in a coordinate system X oriented along the . . ." **Done**

L138: What was the reasoning for picking $C_d^{1/2}\Gamma_{TS}$ as the uncertain model parameter rather than $C_{d}$, the ice-ocean drag coefficient?
**The Stanton number is a turbulent exchange coefficient that determines the exchange of heat across the ice-ocean interface. This is a process that is poorly represented and a source of considerable uncertainty, and although part of that relates to uncertainty in roughness at this interface (hence $C_{d}$), that would alter the rate of heat transfer, all studies of ice shelf melting that I am aware of use the Stanton number or just $\Gamma_{TS}$ as their tuneable parameter to match observed melt rates.**

L142: Remove "of" **Done**

L148: Was there a specific reason you left out melting from geothermal heat flux? It seems like this would be a relatively simple addition to the model since you can assume it stays constant through 2100. I don't think this would be a reason to re-run the simulations, but I am just curious.
**We have expanded on this in the revised manuscript**

L151: Check the formatting of your references here – I think they should be "Pollard et al. (2015) and DeConto et al. (2021)
**This has been fixed here and elsewhere**

L158: General comment about units, make sure to add a space between the value and the unit (e.g.,

1.5 m/yr rather than 1.5m/yr). I saw this come up in a couple of lines now. The only unit that should be directly against the value is the degree-symbol.

**All units should now be correctly formatted**

L174-175: Citation should not have parentheses

**Done**

L200: What version of BedMachine do you use?

We used Bedmachine v3.4 (03-Jun-2022).

**This is now stated explicitly and the reference is correct for this particular version**

L206: What specifically are these assumptions? Do you assume floating ice is at hydrostatic equilibrium? If so, please state this and other assumptions explicitly.

**Yes we assume floating ice is at hydrostatic equilibrium and this is now clarified in the revised manuscript.**

L235-237: This description of how ocean T/S is applied to the plume model is a bit vague. So within each of the three major ice shelf cavities (Pine Island, Thwaites, and Crosson/Dotson), you extract the thermocline depth and T/S above and below that depth. So you get 15 values (5 for each of the three ice shelves), but how is that T and S applied within the plume model? At all ice shelf depths above the thermocline, you apply the average T above the thermocline, and vice versa for below the thermocline? Is this done at every time-step? How does the location where you obtain these T/S/depth variables change as the ice shelf cavity geometry changes in time (i.e., do you always sample from the same locations, or does the location follow migration of the grounding line and ice front)?

Yes, we end up with 15 values for each point in time. Temperature (or salinity) below the thermocline is constant, then from the thermocline depth to z=0 temperature increases linearly to the surface temperature. The temperature, salinity and thermocline depth fields are sampled from the MITgcm model 10 times a year and at each ice sheet model time step we find the closest point in time in the corresponding MITgcm model to extract these 15 values. For all simulations, we define 3 basins for Pine Island, Thwaites and Dotson/Crosson, based on present day drainage basins, defined everywhere in the domain and extrapolated out into the ice shelf, and check which basin a node lies at any point in time to determine which MITgcm field we use for that location.

**We will have expanded the description of this in the revised manuscript and also now point the reader to Appendix A which contains a more complete description.**

L415: It might be nice in this section to also mention the rate of sea level rise of the ASE, not just the final 2100 contribution. In figure-5, it looks like the rate of sea level contribution from RCP8.5 starts to decrease after 2070, do you know why that is?

The rate of sea level contribution for RCP8.5 (perhaps more clearly seen in Fig. 9) is relatively stable until approximately 2060, after which it declines until approximately 2100, before increasing again. We can add a reference to Fig. 9 in this section to point the reader to this information. Regarding what causes this reduction in the rate of sea level contribution, our interpretation is that this is caused by the increase in snow accumulation seen in RCP8.5, acting to counteract dynamic changes in the region and thus reducing the sea level contribution in this period. Once the surface mass balance stabilises (as we remove the trend from 2100 onwards) this effect is removed and continued changes in ice dynamics lead to further adjustment in the regional sea level rise contribution.

**We have added extensive additional analysis and discussion of these points in the revised**

**manuscript.**

L435-455: I agree with the other reviewers that further discussion of these results would be really valuable! In particular, it is quite surprising that you see little grounding line retreat across both Pine Island and Thwaites Glaciers – some analysis on the ice shelf melt rates that were applied throughout these simulations would be very helpful. Perhaps you could show what the melt rates look like at a couple of different time-steps? You could also show a time-series of total integrated ice shelf basal melt across the main shelves in the domain.

Many model outputs were not saved due to the number of simulations involved. We are re-running a representative sample of simulations with increased outputs including melt rates and will add extended analysis including melt rates to the revised manuscript.

**We now include both of the reviewer's suggestions: a plot showing the spatial distribution of melt rates at various points in time and a time series of basal melt rates integrated over each of our model catchments, together with discussion of these points.**

L451 In figure-6, do you know why Pine Island Ice Shelf ends up shaped like a triangle at both 2100 and 2250? I am very surprised to see that this same ice front shape holds throughout the whole simulation.

As mentioned in our reply to reviewer 1 and in the paper, we selected this calving law not necessarily because we believe it to be the most physically plausible, but because we wanted to be able to make some limited comparisons to results of the Pollard and DeConto modelling studies. We find this calving law tends to result in quite consistent calving front positions and in particular for Pine Island Glacier these seem to be relatively insensitive to grounding line position (although there are some examples of very different calving front positions that can be seen upon closer inspection of Fig. 6). In the case of Pine Island Glacier, the calving front seems to be generally anchored to sidewalls where the embayment noticeably widens, and presumably this is primarily driven by the dependence of basal and surface crevasse depth on ice divergence (Eqs B.1a and B.1b in Pollard et al. 2015).

L478: Remove "in the" once; you have it written twice consecutively in this line. **Done**

L481: This comparison to past results is a great starting point! I do think that you should go a step farther and try to deduce why your modeled sea level contributions are on the low end of other published results. Like I said before, I think it is worth looking at what your ice shelf basal melt rates look like in the Ua_fwd simulations and how they change in time (a figure of this would be valuable, maybe in an appendix or supplement). Another possible cause of limited mass loss could be the treatment of surface mass balance, which you could look into by quantifying how much snow fell over the simulation period and how much this offsets ice loss at the grounding line. Do the other projections you cited include atmospheric forcing as a positive degree day parameterization, or do they directly apply CMIP anomalies in SMB within their model without correction?

We are re-running a sample of simulations that will enable us to expand on the analysis, and will investigate these ideas and other excellent suggestions to add to the analysis for the revised manuscript.

**We have included a figure showing the time-evolution of basal melt rates for one simulation until 2300 and we have also included a much more detailed discussion of why we believe our sea level contributions are at the low end of previously published results.**

L495: Is this statement about your melt rates true? I thought that the plume parameterization updates ice shelf melt based on changing ice shelf basal slopes and the depth of the grounding line? Maybe it would be more accurate to say that you cannot resolve 3-dimensional sub-ice shelf ocean circulation.

The statement specifically mentions ocean forcing, which in this case are derived from the un-coupled ocean model simulations, rather than the melt rates calculated by the plume model. The former are not updated in response to changing cavity geometry whereas the latter are, however this is certainly easy to misinterpret and we will reword the sentence to make our meaning clearer.

**We have clarified this point in the revised manuscript.**

L505: Might be worth comparing to the new Science Advances paper by Prof. Morlighem (https://www.science.org/doi/10.1126/sciadv.ado7794). They do not find any evidence for MICI over the next 50 years, so this supports your findings well.

**We have added this citation to the revised manuscript**

L525-530: A similar analysis of results for your 2100 simulations might be helpful in determining why mass loss is on the lower end of published projections.

**We now include a new figure that shows a timeseries of various mass gain and loss terms over the entire simulation period, and have used this to discuss in more detail why our simulations to 2100 are at the low end of published projections**

L533-535: Can you determine or at least speculate on why there is a decrease in RCP8.5 mass loss prior to 2100?

**We have included substantial new analysis to delve into this behaviour and have added discussion around this topic in the revised manuscript. In short, RCP8.5 has the largest increase in snow accumulation between 2021 and 2100, which in our simulations more than offsets other changes in mass balance, leading to a slowing in the SLR contribution.**

L541: The sea level contribution is similar up until 2100, but not through 2250. Please specify this.

**We have clarified this statement pertains to simulations up to 2100 only.**

L560: Check the format of the citation, should have parentheses.

**Done**

Figure Comments:

Figure 1: This is a very helpful figure and I like that you included the associated section and appendix numbers in each of the boxes. In the figure caption, can you please include the definition of acronyms (e.g., RNN, LSTM, and Del GMSL).

**We have included definitions for these acronyms in the revised manuscript.**

Figure 2/3: Should there be a color bar associated with this figure? Also, is the top row showing surface ice speed, or the change in surface ice speed (as would be consistent with a divergent colorbar)? This last comment applies for figure-3 as well – should this say change in surface ice speed? Lastly, for figure-3, why are there no changes shown over floating ice?

We will add a colorbar to these figures and correct any reference to ice velocity that should be surface ice speed. We mask out changes on floating ice shelves and do not include these in the Bayesian inference step since this would complicate our interpretation. As the calving front evolves in our simulations, changes in its location such that the extent of observational data and model data no longer match could create a very strong sensitivity to parameters affecting this process in comparison to other changes. In addition, the surface elevation change dataset only includes data on grounded ice, due to the processing and larger errors involved in calculating this field over floating ice shelves.

**We have added a colorbar and corrected the naming to ensure it is consistent throughout.**

**References:**

Naughten, K. A., Holland, P. R., Dutrieux, P., Kimura, S., Bett, D. T., & Jenkins, A. (2022). Simulated twentieth-century ocean warming in the Amundsen Sea, West Antarctica. *Geophysical Research Letters*, 49, e2021GL094566. https://doi.org/10.1029/2021GL094566

Castleman, B. A., Schlegel, N.-J., Caron, L., Larour, E., and Khazendar, A.: Derivation of bedrock topography measurement requirements for the reduction of uncertainty in ice-sheet model projections of Thwaites Glacier, The Cryosphere, 16, 761–778, https://doi.org/10.5194/tc-16-761-2022, 2022.

Wernecke, A., Edwards, T. L., Holden, P. B., Edwards, N. R., & Cornford, S. L. (2022). Quantifying the impact of bedrock topography uncertainty in Pine Island Glacier projections for this century. *Geophysical Research Letters*, 49, e2021GL096589. https://doi.org/10.1029/2021GL096589

Gudmundsson, G. H., Barnes, J. M., Goldberg, D. N., & Morlighem, M. (2023). Limited impact of Thwaites Ice Shelf on future ice loss from Antarctica. *Geophysical Research Letters*, 50, e2023GL102880. https://doi.org/10.1029/2023GL102880

Pollard, D., DeConto, R. M., and Alley, R. B. (2015) Potential Antarctic Ice Sheet retreat driven by hydrofracturing and ice cliff failure, Earth and Planetary Science Letters, 412, 112–121, https://doi.org/https://doi.org/10.1016/j.epsl.2014.12.035